# Extinction of eastern Sahul megafauna coincides with sustained environmental deterioration

Scott A. Hocknull [1,2✉], Richard Lewis [3], Lee J. Arnold[3], Tim Pietsch[4], Renaud Joannes-Boyau [5], Gilbert J. Price [6], Patrick Moss [6], Rachel Wood[7,8], Anthony Dosseto [9], Julien Louys [10], Jon Olley[4] & Rochelle A. Lawrence [1]

Explanations for the Upper Pleistocene extinction of megafauna from Sahul (Australia and New Guinea) remain unresolved. Extinction hypotheses have advanced climate or human-driven scenarios, in spite of over three quarters of Sahul lacking reliable biogeographic or chronologic data. Here we present new megafauna from north-eastern Australia that suffered extinction sometime after 40,100 (±1700) years ago. Megafauna fossils preserved alongside leaves, seeds, pollen and insects, indicate a sclerophyllous forest with heathy understorey that was home to aquatic and terrestrial carnivorous reptiles and megaherbivores, including the world's largest kangaroo. Megafauna species diversity is greater compared to southern sites of similar age, which is contrary to expectations if extinctions followed proposed migration routes for people across Sahul. Our results do not support rapid or synchronous human-mediated continental-wide extinction, or the proposed timing of peak extinction events. Instead, megafauna extinctions coincide with regionally staggered spatio-temporal deterioration in hydroclimate coupled with sustained environmental change.

[1] Geosciences, Queensland Museum, 122 Gerler Rd., Hendra, QLD 4011, Australia. [2] School of BioSciences, Faculty of Science, University of Melbourne, Melbourne, VIC 3010, Australia. [3] School of Physical Sciences, Environment Institute, and Institute for Photonics and Advanced Sensing (IPAS), University of Adelaide, North Terrace Campus, Adelaide, SA 5005, Australia. [4] Australian Rivers Institute, Griffith University, Brisbane, QLD 4122, Australia. [5] Geoarchaeology and Archaeometry Research Group, Southern Cross GeoScience, Southern Cross University, Lismore, NSW 2480, Australia. [6] School of Earth and Environmental Sciences, The University of Queensland, Brisbane, QLD 4072, Australia. [7] Radiocarbon Facility, Research School of Earth Sciences, Australian National University, Building 142 Mills Road, Canberra, ACT 2601, Australia. [8] School of Archaeology and Anthropology, Australian National University, Building 44, Daley Road, Canberra, ACT 2601, Australia. [9] Wollongong Isotope Geochronology Laboratory, School of Earth, Atmospheric and Life Sciences, University of Wollongong, Wollongong, NSW 2522, Australia. [10] Australian Research Centre for Human Evolution, Environmental Futures Research Institute, Griffith University, Mount Gravatt, QLD 4122, Australia. ✉email: scott.hocknull@qm.qld.gov.au

The Pleistocene megafauna of Sahul are defined here as non-marine vertebrates that exceed a mass of ~40–44 kg[1], comprising giant species of birds, reptiles and marsupials. They have vastly different phylogenetic histories[2], occupying a diverse range of habitats[3–5] and reaching maximal body size by the Upper Pleistocene[6–8] (126,000–11,700 years ago (ka)). In the context of this work we do not consider unusually large-bodied taxa with a mass <40 kg as megafauna, but acknowledge that many species <40 kg also suffered extinction during the Pleistocene[9,10]. Some megafauna species were widely distributed across Sahul during the Quaternary, such as the giant wombat-like *Diprotodon optatum*[7]; however, data on the biochronology and palaeobiogeography of most species remains poor and patchy[9,10].

The megafauna fossil record of Sahul worsens when only those vertebrate fossil sites considered to be reliably-dated are considered[11]. Thus assessing extinction chronologies using the presently available vertebrate fossil record is hampered by few reliably-dated Upper Pleistocene sites within Marine Isotope Stage 3 (MIS 3) (57–29 ka). This period of time is particularly important because it occurs at a time controversially proposed to encompass the complete continental-wide extinction of megafauna[12,13] with the peak of extinction events occurring around 42.1 ka[10,14].

To augment this poor record, dung fungal spores (*Sporormiella*) recovered from lake and marine sediments have been used as a proxy of megafauna presence and abundance with reduced spore counts interpreted as indicating megafauna decline and extinction[15,16]. However the validity of fungal spores as useful proxies for megafauna requires cautious interpretation due to the numerous factors impacting their dispersal and survival into the fossil record, including taphonomic, sedimentological and biological considerations[17,18]. Without a clear link between Sahul megafauna and these dung fungi its utility as a robust proxy is equivocal[10,19].

Current explanations for megafauna extinction are therefore based on a significant spatio-temporal gap in the vertebrate fossil record, which approximates three quarters of the area of the Sahul continent (Fig. 1e). This problem is most evident for central and northern Australia, and New Guinea, where no reliably-dated megafauna occur within MIS 3. This conspicuous gap has not restricted the development of generalised and polarised interpretations to explain the continental-wide extinction of the megafauna, with the prevailing extinction scenarios advancing climate change[9,20] or anthropogenic[12,14,21,22] factors.

The discovery of fossil bones by Barada Barna Traditional Owners at South Walker Creek (SWC), Queensland Museum Locality (QML) 1470, near the township of Nebo, north-eastern Australia, resulted in systematic excavation of new fossil deposits that recovered Pleistocene megafauna. These sites are located in tropical Australia at the northern-most portion of the Fitzroy River Basin (FRB), an east-draining catchment that exits south onto the southern Great Barrier Reef, near Rockhampton, central-eastern Queensland (Fig. 1). Importantly, the FRB is located within the 'data gap' for megafauna and adjoins Australia's two largest basins to the west. Both of these basins, the inland-draining Lake Eyre Basin (LEB) and the south-draining Murray-Darling Basin (MDB), have reliably-dated megafauna sites and possess established palaeoenvironmental proxies[20,23–28].

Here we present a new and diverse Pleistocene megafaunal assemblage unique to eastern Sahul. We then provide a chronology for these megafauna that combine observations of site geology and taphonomy with a chronometric dating assessment. Reliable numerical ages are reported for four megafauna sites using multiple dating techniques and applied quality rating criteria[11]. These results are then integrated into the established regional and local palaeoenvironmental and archaeological records that provide important context for testing the prevailing megafauna extinction hypotheses for Sahul. Our results do not support rapid or synchronous human-mediated continental-wide extinction or the proposed timing of peak extinctions around 42.1 ka. Instead, megafauna extinctions coincide with regionally staggered spatio-temporal deterioration in hydroclimate coupled with locally sustained environmental changes that were arguably detrimental to megafauna survival.

## Results

**New and diverse tropical megafauna.** Four sites along a stretch of Walker Creek represent the South Walker Creek fossil deposits reported here that contain the remains of a new and diverse megafauna assemblage (Fig. 2). Fossil fauna and flora are taxonomically identified along with descriptions of these remains (Supplementary Notes 1–3), their taphonomic, depositional and stratigraphic context (Supplementary Notes 2 & 4). The deposits preserve body and trace fossils from small to megafauna-sized vertebrates, aquatic and terrestrial invertebrates (e.g. bivalves and insects) and floral remains (e.g. seeds, leaves and pollen) (Fig. 2a, f–x, Supplementary Figs. 1–7 & 10).

The combination of fauna and flora preserved within an open fossil site in Australia is rare. Recently, modelled suitability ratings for preservation and discoverability of megafauna fossils rates the SWC area at, or close to, zero[29]. This suggests that either these fossil deposits are truly exceptional or the models require further refinement before being considered useful. The SWC sites thus offer a rare opportunity to develop a holistic account of megafauna within a tropical northern palaeoenvironment from northern Australia and eastern Sahul.

At least sixteen megafauna species are present, including thirteen extinct and three extant species. Carnivorous megafauna are represented by the marsupial 'lion', *Thylacoleo*, at least three crocodilians (*Crocodylus* sp. cf. *C. porosus*, *Pallimnarchus* sp. and 'Quinkana' sp.) and two giant monitor lizards (*Varanus priscus* ("Megalania") and a Komodo Dragon-sized species). Mega- and Macroherbivores are diverse with five species of kangaroo, two wombats, one palorchestid, two diprotodontids and the emu, *Dromaius* sp. cf. *D. novaehollandiae* (Table 1, Supplementary Note 1).

The macropodids are morphologically distinct taxa but remain largely enigmatic due to low numbers of preserved dental remains and uncertain species-level taxonomies (Supplementary Note 1 and Supplementary Fig. 5). Five species of macropodine, including a species each of *Macropus*, *Notamacropus*, *Osphranter* and *Protemnodon*, as well as a species of small-sized sthenurine, define a kangaroo fauna that bears little resemblance to the 'typical' Upper Pleistocene sthenurine-dominated kangaroo faunas of southern Australia[6,30]. Also, macropodines that are commonly documented from Pleistocene open sites in the bordering MDB catchment to the south of the FRB; such as *Macropus giganteus titan*, *Macropus ferragus*, *Notamacropus agilis siva* and *Protemnodon anak*[31], are notably absent from the SWC macropodid fauna. These taxa are of similar body size to those found at SWC; therefore, it is unlikely that these faunal absences are due to taphonomic sampling bias.

The SWC macropodines are all high crowned forms, including the species of *Protemnodon*, which is either a new species or a northern variant of the similar-sized *Protemnodon brehus* (Supplementary Note 1; Supplementary Fig. 5, ab–an). The largest and most common macropodine is possibly a new species of *Macropus* that possesses a combination of characteristics found in Pliocene-aged *Macropus pan* and Pleistocene-aged *Macropus pearsoni*. Comparison of limb dimensions of this giant species to

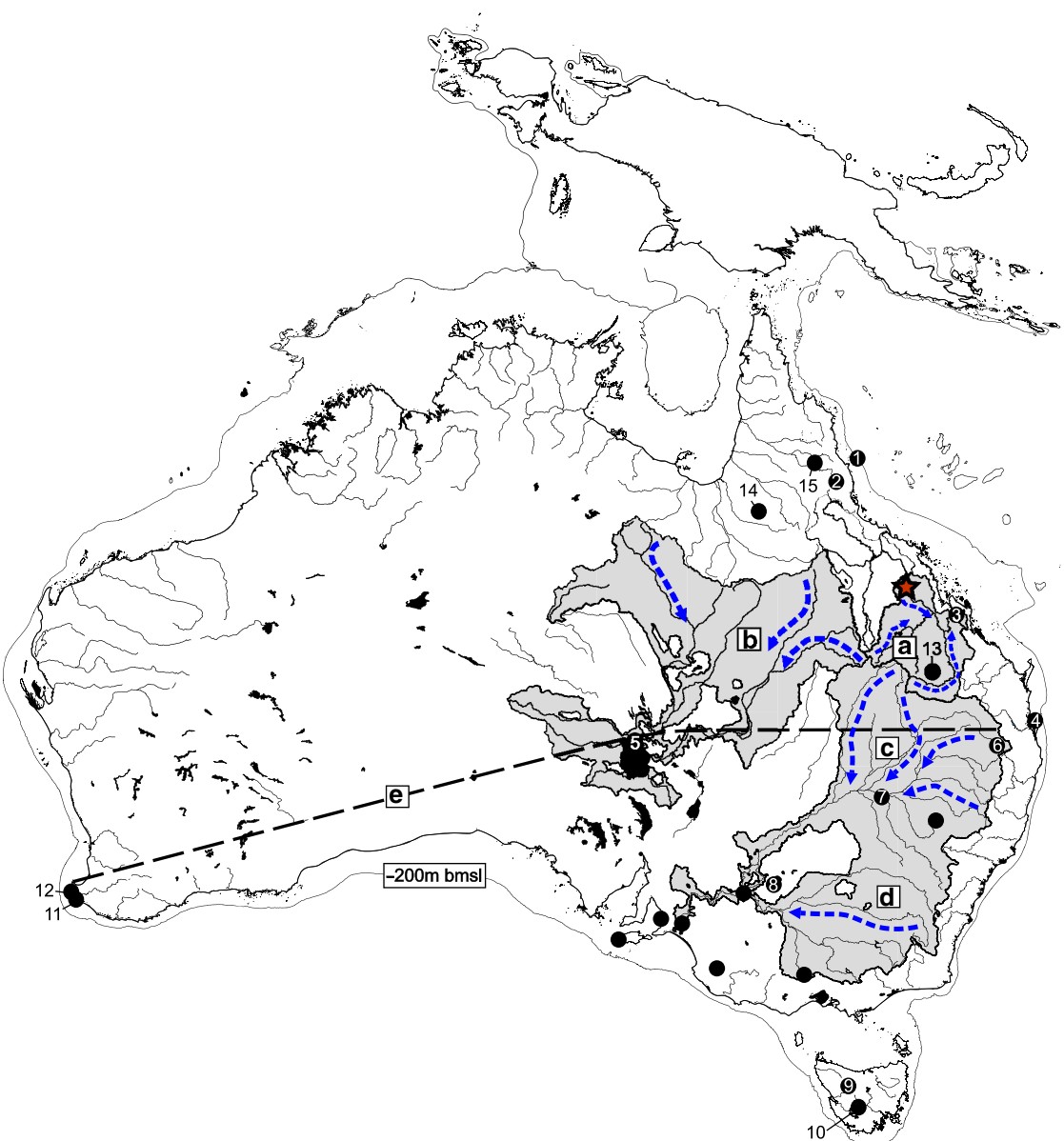

**Fig. 1 Map of Sahul (Australia and New Guinea) showing the distribution of reliably-dated megafauna sites within MIS 3 (57–29 ka) and locations mentioned in the text.** Red star indicates South Walker Creek. **a** The Fitzroy River Basin (FRB). **b** The Lake Eyre Basin (LEB). **c** The northern Darling and **d** southern Murray River catchments of the Murray-Darling Basin (MDB). **e** Over three quarters of the continental area of Sahul is missing reliably-dated sites from MIS 3, here indicated north of the dashed line. Other localities mentioned in text, 1. ODP 820, 2. Lynch's Crater, 3. Capricorn Caves, 4. North Stradbroke Island, 5. Kati Thanda–Lake Eyre, 6. Ned's Gully, 7. Cuddie Springs, 8. Lake Mungo, 9. Mt. Cripps, 10. Titan's Shelter, 11. Tight Entrance Cave, 12. Kudjal Yolgah Cave, 13. Kenniff Cave, 14. Gledswood Shelter and 15. Ngarrabullgan. Blue arrows indicate catchment flow direction. bmsl = below mean present day sea level indicating outline of the Sahul continent. The baseline map was generated in QGIS using shoreline data from https://www.ngdc.noaa. gov/mgg/shorelines/ under GNU Lesser General Public License v3 or later; and drainage basin data[111] from https://data.gov.au/data/dataset/f55ec9b3-ab74-4056-93a2-b4b8aa65ead1 under Creative Commons Attribution 4.0 International; and bathymetry data[112] from https://data.gov.au/data/dataset/ australian-bathymetry-and-topography-grid-june-2009 under Creative Commons Attribution 4.0 International. The base map was composited using Corel Draw and altered to delineate 'Lake Carpentaria'.

those of other giant macropodid skeletons indicates that this taxon is the largest species of kangaroo. Maximum tibial length is at least 250 mm longer than that of the largest known specimens of *Procoptodon goliah*, *Macropus giganteus titan*, and *Macropus ferragus*[32] (Supplementary Note 1; Supplementary Figs. 4 & 5, l-aa). Based on femoral circumference, mass is estimated at around 274 kg, ~43 kg heavier than the second largest species, the sthenurine *Procoptodon goliah*[33] (Supplementary Note 3). A species of *Notamacropus* is distinguishable from *N. agilis siva*, thus likely a new species or another northern variant

(Supplementary Note 1; Supplementary Fig. 5a–i). A species of *Osphranter* is morphologically indistinguishable from very large individuals of the extant Red Kangaroo (*Osphranter rufus*) (Supplementary Note 1; Supplementary Fig. 5j). Together, the dominance of new macropodid forms along with the absence of common and widely distributed species indicates that the Pleistocene palaeoenvironment at SWC is presently unique in Sahul.

A species of *Palorchestes* is morphologically similar to *P. azael* but much smaller and is distinguishable from species similar in

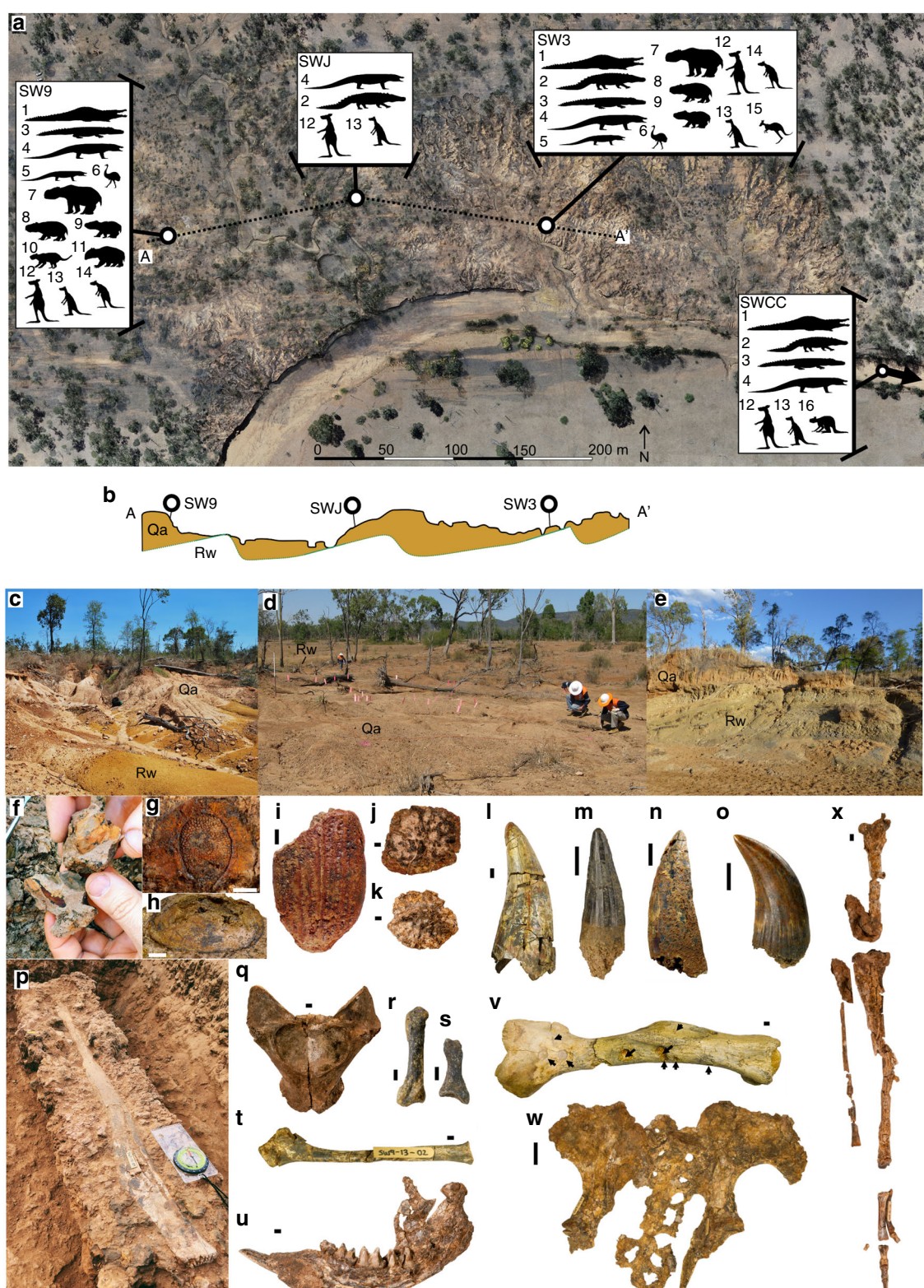

size (e.g. *P. parvus*) (Supplementary Note 1; Supplementary Fig. 6a–c). This species may represent either a new species or another northern variant. The remaining megafauna are referrable to known taxa, including *Diprotodon optatum*, *Zygomaturus trilobus*, *Phascolonus gigas* and *Sedophascolomys* sp. cf. *S. medius* (Supplementary Note 1; Supplementary Figs. 6 & 7). The specimens of SWC *Diprotodon optatum* are within the

smallest size range when compared with those of Pleistocene sites from the MDB and LEB[7,34] (Supplementary Table 2).

**Reliable megafauna ages for eastern Sahul.** The four SWC sites presented here were assessed geologically, taphonomically and chronometrically so that the new fauna could be placed within a

**Fig. 2 Summary of field sites and diversity of fossil remains from South Walker Creek (QML1420). a** Aerial map of main South Walker Creek fossil sites SW9, SWJ and SW3 with downstream site SWCC indicated by arrow. Megafauna taxa recovered from each site indicated by numbered silhouette: 1. *Pallimnarchus* sp. 2. '*Quinkana*' sp., 3. *Crocodylus* sp. cf. *C. porosus*, 4. *Varanus priscus*, 5. *Varanus* sp. (large), 6. *Dromaius* sp., 7. *Diprotodon optatum*, 8. *Phascolonus gigas*, 9. *Sedophascolomys* sp. cf. *S. medius*, 10. *Thylacoleo* sp., 11. *Palorchestes* sp., 12. *Macropus* sp. (giant), 13. *Protemnodon* sp., 14. *Notamacropus* sp. (giant), 15. *Osphranter* sp. cf. *O. rufus*, 16. sthenurine. **b** Stratigraphic section A–A′ crossing through SW9, SWJ and SW3 (indicated by dashed line in (**a**)). Quaternary (Qa) alluvial sediment overlies dipping basement Permo-Triassic Rewan Group (Rw) bedrock (vertical exaggeration 5×). Fossil deposit surface expression at SW9 (**c**), SW3 (**d**) and SWCC (**e**). Summary of the diverse fossil remains recovered from SWC sites (see Supplementary Note 1 for detailed descriptions): **f** leaves and bivalves in situ at SW9; **g** Goodeniaceae seed; **h** *Velesunio wilsoni* bivalve; **i** insect elytron (?Curculionidae); **j** *Pallimnarchus* sp. osteoderm; **k** *Crocodylus* sp. cf. *C. porosus* osteoderm; isolated crocodile teeth from **l** *Pallimnarchus*, **m** *Crocodylus* sp. and **n** '*Quinkana*'; **o** *Varanus priscus* tooth; **p** *Macropus* sp. (giant) tibia in situ at SW9; **q** *Varanus priscus* dorsal vertebra; associated appendicular elements from *Thylacoleo* sp., **r** metacarpal, **s** phalange and **t** fibula; **u** *Diprotodon optatum* right dentary; **v** *Macropus* sp. (giant) humerus with crocodile puncture marks (indicated by arrows); **w** Articulated pelvis and caudal vertebrae of *Phascolonus gigas* from SW9; **x** associated hind limb of juvenile *Protemnodon* sp. from SW9. Scale bars equal 1 mm in (**g**, **i**); 5 mm in (**h**, **j**–**t**, **v**, **x**); 10 mm in (**u**); and 50 mm in (**w**).

reliable chronology. The detailed results of these assessments are provided in Supplementary Notes 3–8. All of the sites occur within Quaternary alluvial deposits accumulated atop shallow-dipping Permo-Triassic bedrock. Each deposit represents a different sedimentary accumulation ranging from small-scale rapid burial events to larger-scale accretionary deposits (Fig. 2a–e; Supplementary Note 4; Supplementary Figs. 11–14 and Supplementary Tables 4–7). Although common, large channel-fill deposits are consistently unfossiliferous.

Taphonomic and sedimentological evidence indicates a primary depositional context for the fossils with no evidence of secondary reworking other than present day exposure and erosion. Taphonomic characteristics of the vertebrate remains range from direct bone modification made by predators (e.g. crocodiles (Fig. 2v) and *Thylacoleo* sp. bite marks (Supplementary Fig. 10d)) to dry and wet bone fractures dominated by weathering stage 0, indicating short-term exposure[35] (Supplementary Note 4; Supplementary Fig. 9). Voorhies Groups I, II and III are observed in combination at most sites and each site preserves different degrees of skeletal association ranging from articulated to isolated elements indicating varying levels of transport[36] (Fig. 2w; Supplementary Note 2 & 4; Supplementary Figs. 5, 6, 9, 10 and Supplementary Table 3). We acknowledge the complex taphonomic processes at play within fluvial systems[37]; however, the presence of articulated and associated remains with limited pre-depositional weathering at two sites in particular (QML1470 (SW9) and QML1470 (SW3)) demonstrates low levels of transport prior to burial (Fig. 2w; Supplementary Note 2 & 4). The preservation of delicate remains, such as insect and seed parts, further supports rapid deposition at SW9 (Fig. 2f–I; Supplementary Fig. 1). Post-depositional alterations of the fossils (e.g. bones, teeth, leaves, seeds, molluscs and insects) are localised at each site and include iron-oxide and carbonate precipitation, cortical bone and tooth enamel splitting, microscopic surface bone striations and bone deformation (Supplementary Fig. 10). Taphonomic characteristics developed for vertebrates[38] form the basis for our interpretation, including subsurface alterations via long-term drying and compaction of the mud-dominated host matrix (Supplementary Note 4). Post-depositional introduction of younger sediment is only observed within the north-east portion at one site (SW9) as sediment-filled cracks. This indicates post-depositional drying of the deposit and introduction of younger sediment into these cracks, which were avoided or sampled separately during sampling for optically stimulated luminescence (OSL) (Supplementary Figs. 11 & 15). Sediment-filled cracks were not observed anywhere else within SW9, or at any other site.

No evidence has been found to indicate reworking of older faunal remains into deposits at SW9, which is independently supported by uranium-series (U-series) assessments of fossilised teeth from SW9 (Supplementary Note 7). Six megafauna species

from SW9 were directly dated using U-series laser ablation and micro-profiling, all returning consistent and younger ages than other techniques, therefore, supporting the primary context of the deposit through the absence of reworked older fossils (Table 2, Supplementary Note 7). Together, this gives us confidence that the chronometric assessment of the fossils and sediments entraining them accurately reflect the depositional age of the fossils.

To chronometrically constrain the age of the megafauna sites we used multiple dating techniques to age the fossils or the sediment they were buried in. Direct dating of fossils is considered the most reliable approach for establishing extinction chronologies with a quality rating of A* or A[11], therefore, we applied radiocarbon, U-series, and U-series combined with electron spin resonance (ESR) techniques to fossilised megafauna teeth and bones from the best preserved and most fossiliferous site, SW9. U-series and U-series-ESR techniques provided viable results (Table 2, Supplementary Notes 7 & 8); however, radio-carbon assessment of megafauna bones was unsuccessful due to a lack of collagen (Supplementary Note 6).

Dating of sediments derived from above, within, and below the fossil-bearing layer using single-grain OSL is considered an A-rated method for indirectly dating fossils and is readily applicable to undisturbed sediment[11]. We applied the OSL dating technique to all of our fossil sites and these returned viable results (Table 2, Supplementary Note 5). The combined geological, taphonomic, faunal, floral and chronometric studies support an undisturbed primary context for the fossil deposits and their remains, thus by combining A* or A-rated dating techniques and using multiple independent laboratories to do so we are confident that the chronometric ages obtained for each site are of high quality and thus reliable[11] (Table 2, Methods and Supplementary Notes 1–9).

The chronometric ages for the four sites are summarised in Table 2 with detailed methodology and datasets provided in the Methods section and Supplementary Notes 1–9. Together, the four sites span ~20,000 years from ~60 to ~40 ka. The oldest fossil-bearing layer occurs at SWJ, which has a maximum weighted mean OSL age of $65.6 \pm 2.2$ ka ($n = 6$). The next youngest fossil layers occur at SWCC, which has a weighted mean OSL age of $58.2 \pm 6.1$ ka ($n = 2$), and SW3 which has a weighted mean OSL age of $47.7 \pm 3.2$ ka ($n = 5$). The youngest fossil deposit which occurs at SW9 has a weighted mean age of $40.1 \pm 1.7$ ka ($n = 24$), which has been calculated from 19 OSL ages and 5 US-ESR ages (Tables 1 & 2, Supplementary Notes 5–8).

**Extinctions coincide with environmental deterioration.** The known limitations of the fossil record, such as those described by the Signor-Lipps effect[39], can bias the estimated timing of extinction for a fossil taxon[40]. Therefore, the youngest fossil record for a particular taxon should only be considered as

**Table 1 Megafauna species within MIS 3 from South Walker Creek and other Fitzroy River basin sites compared with species lists from other similarly-aged regions of Australia.**

| Region | | TAS | SW-WA | Lake Eyre Basin | Murray-Darling Basin | | | Fitzroy River Basin | | | | Other sites |
|---|---|---|---|---|---|---|---|---|---|---|---|---|
| Sites | | | | | Willandra Lakes | Ned's Gully | Cuddie Springs SU6A + B | SW9 | SW3 | SWCC | SWJ | |
| Chronometric ages (ka) | | 48– ~41 | 50–41 ± 2 | 47.5 ± 2.5 | >33– <36 | 47 ± 4 | ~35 ka– >45–~50 | 40.1 ± 1.7 | 47.0 ± 3.2 | 58.2 ± 6.1 | <65.6 ± 2.2 a | <50 |
| A or A* ages in bold | | n = 12 | n = 12 | n = 11 | n = 3 | n = 2 | n = 5 | n = 24 | n = 5 | n = 2 | n = 6 | N/A |
| Dating Criteria rating[11] | | A–A* | A | A + A* | A + B | A | A* + B | A + A* | A | A | A | |
| †Progura (egg shell) | MeH | | | – | | | | | | | | |
| **Megafauna** | | | | | | | | | | | | |
| †Genyornis | McH | | | | | | | | | | | |
| Dromaius sp. | McH | – | – | – | | | – | Y | – | – | – | |
| †Pallimnarchus sp. | MC | | – | | | | – | Y | – | – | – | b |
| Crocodylus sp. cf. C. porosus | MC | | | | | | | Y | – | – | | |
| †'Quinkana' sp. | MC | | | | | | | | – | – | – | c |
| †Varanus sp. (large) | LC | | | | | | | Y | – | – | – | |
| †Varanus priscus | MC | | | | | | | Y | – | | | |
| Macropus fuliginosus | MeH | | – | | | | | | | | | |
| Macropus giganteus | MeH | | – | | | | – | | | | | |
| †Macropus titan | McH | Y | – | | | Y | | | | | | |
| †Macropus sp. (giant) | McH | | – | | | | | Y | – | – | – | |
| †Notamacropus sp. (giant) | McH | | – | | | | | Y | – | – | | b |
| Osphranter rufus | MeH | | | | | | – | | – | | | |
| †Protemnodon sp. | McH | Y | Y | | | d | – | Y | – | – | | |
| †Sthenurus sp. | McH | | | | | d | – | Y | | | | |
| †Procoptodon sp. | McH | | Y | | | | | | | | | |
| †Simosthenurus sp. | MeH | – | 2spp | | | d | – | Y | | | | |
| †Phascolonus gigas | McH | – | | | | d | – | Y | – | – | | |
| †Sedophascolomys sp. cf. S. medius | MeH | | Y | | | | | Y | Y | | | b |
| †Diprotodon optatum | MH | – | – | | | Y | – | Y | – | – | | b |
| †Zygomaturus trilobus | MH | | | | ?Y | | | Y | | | | ?b |
| †Palorchestes sp. | McH | | | | | d | | Y | | | | |
| †Thylacoleo sp. | MC | – | – | – | | d | | Y | | | | |
| Extinct Megafauna | | 3 | 7 | 0 | | 7 (+2 uncertain) | | 13 (+1 uncertain) | | 3 | | |
| Extant Megafauna | | 2 | 3 | 1 | | 3 | | | | | | |
| Total Megafauna species | | 5 | 10 | 1 | | 10 +2 uncertain | | 16 +1 uncertain | | 3 | | |

Megafauna listed are from sites within the last 50 ka that have been reliably-dated with an A or A* rating using defined rating criteria[11] (see Methods). Those with equivocal B-ratings are discussed in the Methods. Y denotes the youngest reliably-dated megafauna from each region. Megacarnivore (MC) (>100 kg)[113], Large carnivore (LC) (21.5–99 kg)[113], Megaherbivore (MH) (>1000 kg)[114], Macroherbivore (McH) (100–1000 kg)[114], Mesoherbivore (MeH) (10–100 kg)[114]. TAS Tasmania, SW-WA Southwest Western Australia, spp species, ka thousands of years.

†Extinct.

aOldest site at SWC (SWJ) included for completeness.

bUndated Upper Pleistocene fauna from Quaternary sites (Kemmis and Homevale) close to SWC included for completeness.

cDated occurrence (30–<50 ka) of a large-bodied varanid from Capricorn Caves, FRB[115], B-rated[116].

dTaxa from Ned's Gully locality without direct association with dated fossil layer[108], B-rated.

**Table 2 Chronometric ages for South Walker Creek sites and their reliability score.**

| QML1470 | Dating method | Association | Score | n | Age interpretation | Age (ka) |
|---|---|---|---|---|---|---|
| SW9 | US-ESR (SCU) | Direct (m*) | Yes—w | A* | 5 | Age of megafauna fossils within Unit C. | 32.5 ± 2.5 (2σ)[a] |
| | US (UQ) | Direct (m) | Yes—w | A | 4 | Minimum age of megafauna fossils within Unit C. | 24.5 ± 0.1 (2σ)[b] |
| | US (UOW) | Direct (m) | Yes—w | A | 2 | Minimum age of megafauna fossils within Unit C. | 25.5 ± 0.1 (2σ)[b] |
| | OSL (AU+GU) | Indirect (m) | Yes—a, w+b | A | 19 | Age of Unit C, including fossiliferous layer. | 41.3 ± 1.9 (1σ)[a] |
| | OSL+US-ESR (ALL) | Both | Yes—a, w+b | A*+A | 24 | Combined age of megafauna and deposit. | 40.1 ± 1.7 (1σ)[a] |
| | C$^{14}$ (charcoal) (ANU) | Indirect (B) | Yes—w+b | B | 3 | Minimum maximum age of deposit. | >40–50 ka[c] |
| SW3 | OSL (AU) | Indirect (m) | Yes—w+b | A | 5 | Age of entire deposit including fossiliferous layer. | 47.7 ± 3.2 (1σ)[a] |
| SWCC | OSL (AU) | Indirect (m) | Yes—a+w | A | 2 | Age of entire deposit (Units B, C & D), including fossiliferous layers. | 58.2 ± 6.1 (1σ)[a] |
| SWJ | OSL (AU) | Indirect (m) | Yes—b | A | 6 | Age of Unit B, maximum age of fossiliferous Unit C. | 65.6 ± 2.2 (1σ)[a] |

Ratings of dating methods uses established criteria[11]. All chronometric results, including detailed methods, are provided in the Methods and Supplementary Notes 5–8. m, m*, B, A and A* are categories and ratings defined here[11].
a above, b below, w within the fossiliferous layer.
[a]Weighted mean age and uncertainty.
[b]Oldest age and uncertainty.
[c]Reworked macro-charcoal represents a maximum age for the deposit, however, this material only survived ABA treatment, therefore must be considered a minimum age range. Radiocarbon dates on sediment are not included here as the data is not acceptable according to Rodrigues-Rey[11].

indicative of the last appearance of an extinct taxon, not its true time of extinction. With this in mind, we can conclude that the extinct megafauna taxa at SWC suffered local extinction sometime after their youngest last appearance age at SW9, which is 40.1 ± 1.7 ka.

With our new fauna and chronology in mind, we investigated available palaeoenvironmental proxies, in particular those recording regional and local changes that are likely to have impacted the survival of megafauna (e.g. water, vegetation, fire and climate). Hydroclimatic proxies that include palaeomonsoon activity, catchment flow and lake levels are available for northern Australia and the three basins associated with our fauna, the FRB, LEB and MDB. Chronologically staggered and abrupt hydroclimatic deterioration starts around 48 ka in the LEB[20,28] (Fig. 3a), then from 45 ka in the MDB[25,26,28] (Fig. 3b and c), and then from 40 ka in the FRB[24,28] (Fig. 3d). Hydroclimate in northern Australia falls below present day levels after 40 ka[28,41] indicating a significant drying trend (Fig. 3e). Together, these records show abrupt and major fluctuations with a sustained reduction in effective rainfall reaching the watersheds of the LEB, MDB and FRB from 48 ka and lasting until at least 30 ka. Within each basin, the youngest reliably-dated megafauna occur during, or just prior to, the onset of abrupt hydroclimatic deterioration (Fig. 3a–e). Given that the extinction of these megafauna occurred after their last appearances, we infer that local extinctions in these basins occurred during abrupt deterioration in hydroclimate.

The correspondence between the staggered chronologies of both the youngest megafauna and the onset of sustained hydroclimatic deterioration for each basin suggests hydroclimatic change as a possible driver of catchment-based extinctions, occurring first in the LEB, then in the MDB and then in the FRB. The sustained loss of effective catchment flow within the FRB would not favour survival of aquatically-dependent megafauna, such as the three crocodilians. *Crocodylus* likely survived this inland deterioration by occupying viable coastal environments whilst *Pallimnarchus* and '*Quinkana*' did not. At some point since 30 ka the hydroclimate of the FRB returned to levels similar to that experienced prior to 40 ka, presumably after the Last Glacial Maximum; however, the aquatic environments that were once conducive for crocodilians at SWC did not return indicating that the hydrological system was irrevocably altered.

We acknowledge that for open sites there may be reduced preservation potential of vertebrate remains due to the deterioration of the hydroclimate; therefore, a taphonomic bias against the preservation of younger sites may have occurred as the hydroclimate deteriorated. This potential bias of preservation can be reconciled with the discovery of reliably-dated sites within these basins that preserve only extant megafauna.

In addition to hydroclimate, we investigated Quaternary proxy records of vegetation[23,27,42], fire[15,27,42,43] and climate change[24,44] from east of the Great Dividing Range in Queensland. These records represent palaeoenvironmental proxies closest to our sites and likely better reflect more direct environmental changes influencing megafaunal survival. Palynological data from the offshore ODP820[27] drill record is the most appropriate for indicating changes in vegetation and fire frequency relevant to our locality because it preserves a signal that is not constrained by local terrestrial site depositional biases[45]. This record indicates an overall drying with sustained reduction of complex rainforest beginning ~50 ka. Importantly, there is a regional reduction in grasslands (sclerophyll herbs), which is a key dietary component of herbivorous megafauna. This reduction begins around 50 ka and reaches its lowest levels for the entire record at ~35 ka (Fig. 3f). Arboreal sclerophyll taxa (e.g. *Eucalyptus*, *Melaleuca*, *Acacia* and Casuarinaceae) generally follow the same trends seen in sclerophyll herbs over the last 250 kyr[23]; however, they

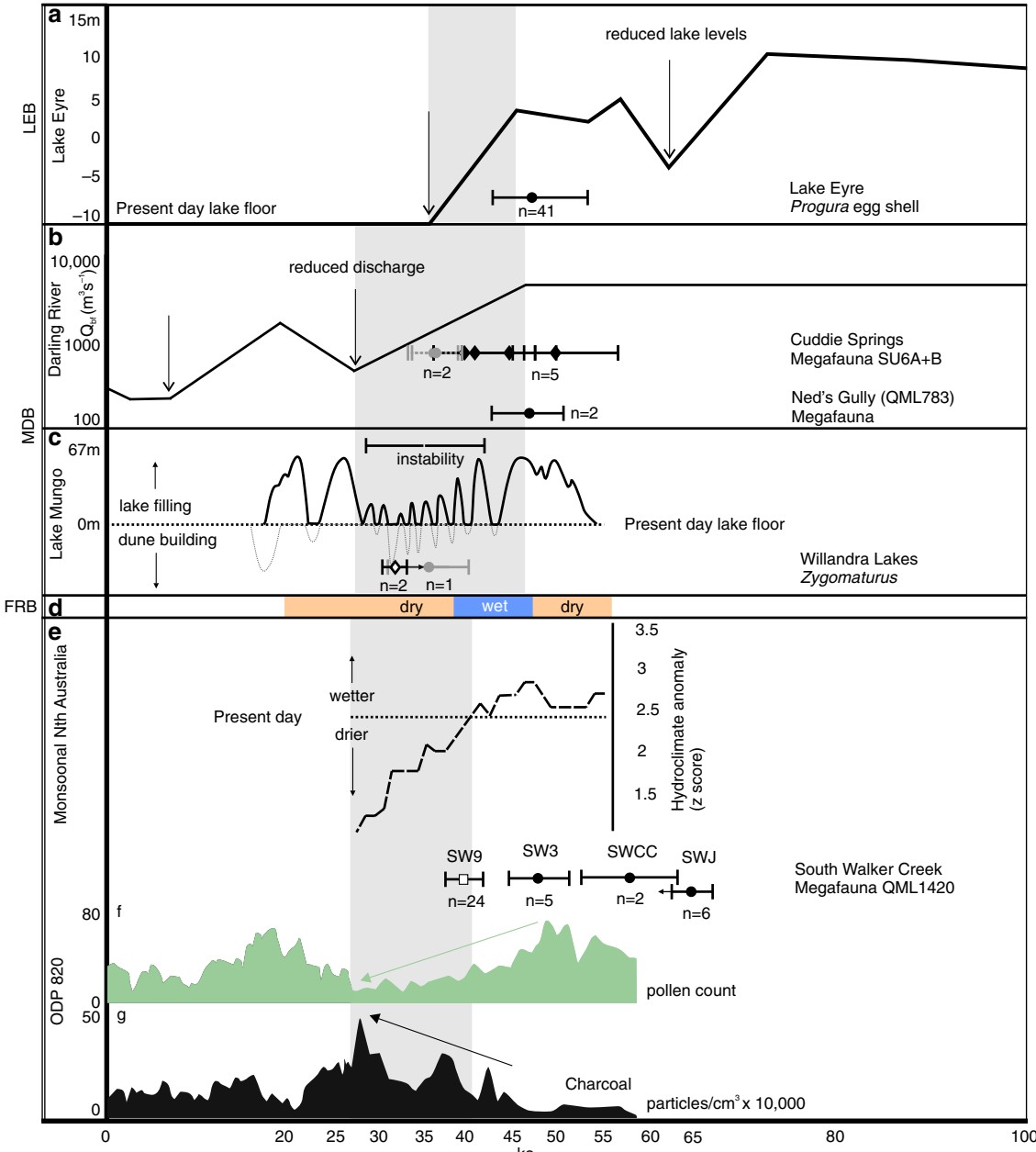

**Fig. 3 Regional hydroclimate and local palaeoenvironmental proxies since 100 ka compared with young megafauna sites from the Lake Eyre, Murray-Darling and Fitzroy River basins. a** Hydroclimatic proxies include Kati Thanda–Lake Eyre levels (adapted from ref. [20]). **b** Palaeodischarge of the Darling River (adapted from ref. [26]). **c** Mungo Lake levels and dune development (adapted from ref. [25]). **d** FRB catchment activity (adapted from refs. [24, 28]). **e** Hydroclimate relative to present day of northern monsoonal Australia (adapted from ref. [28]). **f** Local palaeoenvironmental proxies including counts of grass pollen (Poaceae) (**f**) and micro-charcoal (**g**) from ODP 820 (adapted from ref. [27]). Young megafauna sites within each basin are plotted using A or A* quality-rated ages (black plots) (Table 1). All SWC sites are plotted here for completeness. Published OSL ages for Cuddie Springs[55] and the Willandra Lakes *Zygomaturus*[107] are B-rated here but are equivocal so we have included them (plotted in grey). Extinctions are interpreted as occurring sometime after the youngest A or A* age. Note the timing of onset of trajectory of hydroclimatic deterioration (light grey bars) within each region and megafauna ages. Closed circles denote OSL ages (1σ); open diamonds denote U-series minimum ages (2σ); filled diamonds denote US-ESR ages (2σ); open circles denote combined radiocarbon and OSL ages (*Progura* egg shell) (1σ); and open square denotes combined OSL and US-ESR ages (1σ).

approximately double their proportional representation after ~36–35 ka[23,27,42] indicating a sustained change in the sclerophyllous vegetation structure along the eastern seaboard.

Our palynological assessment of SW9 shows similarities to the regional pollen record of ODP820 and the local record at Lynch's Crater[46] during MIS 3. The overall taxonomic composition of pollen taxa at SW9 is also similar to that of the present day SWC area (Supplementary Note 9, Supplementary Fig. 34 and Supplementary Table 23), therefore, this indicates limited change in floral composition since the deposition of SW9 ~40 ka. A greater proportional representation of arboreal sclerophyllous taxa compared with low counts of sclerophyllous herbs (e.g. Poaceae) is surprising but may be indicative of the observed regional reduction of grasslands from 50 to 35 ka and suggests

suboptimal forage for the grazing megafauna. The dominance of arboreal sclerophyllous taxa may also be indicative of the transition to a more dominant arboreal sclerophyll vegetation structure recorded later in MIS 3 further to the north[23,27,46].

The ODP820 charcoal record illustrates an increased fire frequency starting ~44 ka, intensifies from ~40 ka, and peaking ~28 ka[23,27,43] (Fig. 3g). While anthropogenic landscape burning has been attributed to these increases over the last 50,000 years[27,43], this has been challenged and a more likely explanation involves the complex relationship between climate and vegetation with fire frequency[47]. Importantly, the increased and sustained burning from ~44 ka indicates a fundamental shift toward environmental deterioration and instability that would have impacted the survival of megafauna. The extinction of grazing megafauna sometime after ~40 ka would have reduced grazing pressure on grasslands and potentially created the conditions favouring intensified burning[48,49], and possibly contributing to the peak in burning seen regionally around 28 ka. Our assessment of charcoal particles at SW9 (Supplementary Note 9) demonstrates preservable indicators of fire within the environment; however, these results will be uninformative until a longer sequence is available to assess and compare relative charcoal frequencies through time as well as any taphonomic bias within this alluvial system.

Together, the multiple proxies investigated here show regionally sustained vegetation change with reduction of grasslands and a sustained higher representation of arboreal sclerophyll taxa, which is also reflected locally at SWC. These vegetation changes occurred alongside sustained and intensified fire along with reduced catchment flow and thus available free water. Such deterioration of the environment is set within a backdrop of intensifying aridity[28]. The culmination of these changes would have impacted terrestrial megafauna survival, especially the herbivores and more specifically the grazers. These negative impacts from a deteriorating environment on herbivorous megafauna survival would likewise have affected the hypercarnivorous megafauna.

The SWC megafauna were a diverse aquatic and terrestrial megafauna community, occurring locally from at least 60 ka until 40.1 ± 1.7 ka, with their local extinction occurring sometime after this. The regional environmental changes that began ~48–50 ka did not have an obvious or immediate impact. However, we propose that the intensification of environmental deterioration and instability starting ~40–41 ka played a key role in their local extinction (Fig. 3).

Reduction of the *Sporormiella* dung fungus occurring around 41 ka at Lynch's Crater, ~560 km north of SWC, has been used as a proxy for both the presence of megaherbivores and their rapid population decline to extinction[15,16]. However, the use of *Sporormiella* as a megaherbivore proxy has been challenged on taphonomic and biological grounds[10,18,19,50,51]. Variable concentrations of *Sporormiella* spores are present in the sediment samples taken at SW9 for palynological assessment (Supplementary Note 9, Supplementary Table 23). Some samples indicate high concentrations similar to those recovered from Lynch's Crater[15] and off-shore Western Australia[16]; however, it is unclear whether there are other depositional or taphonomic biases at play in the fluvial system that can explain such variation in concentrations. Therefore, the presence of some high concentrations of *Sporormiella* potentially supports the presence of megaherbivores within the environment at the time of SW9 deposition; however, this will remain equivocal until a detailed assessment of depositional and taphonomic biases is undertaken.

A role for people in the extinction of Sahul megafauna through their direct extirpation has been previously proposed. However, with no evidence of butchery or kill sites, it has been proposed

that extinction occurred rapidly across Sahul shortly after human arrival[12–14,22,52–55]. In the absence of evidence for direct extirpation, indirect human-mediated factors such as landscape burning have been proposed but are difficult to differentiate from non-human factors[15,16,21,56,57].

The occupation of Sahul most-likely occurred in northwest Sahul, at least 50 ka[58,59] or as early as 65 ka[60], with much of mainland Australia inhabited by 45 ka[61–63]. People most-likely dispersed across the continent via migratory routes either reflecting connectivity to available freshwater[64] or first along coastal routes and then dispersing inland[65]. Therefore, under a human-driven extinction scenario where megafauna suffer extinction shortly after human arrival it would be expected that megafauna would suffer extinction first in the north and then along the continental periphery of Sahul before extinction occurred in southern and interior regions.

SWC represents the most northerly and the only diverse megafauna site from the easterly-draining seaboard to be reliably-dated within the period of human occupation of Sahul. Therefore, it is well situated to test these scenarios. If the youngest megafauna ages in an area are considered to be the closest age approximation for their extinction, then the most reliable records indicate that extinctions occurred first within the interior (e.g. LEB) after 48 ka, then within the southern MDB after 45 ka; and then around the continental periphery (e.g. southwest Western Australia, Tasmania and SWC) after 41–40 ka (Table 1). This result is contrary to what would be expected and so our assessment does not support rapid or synchronous continental-wide extinction of megafauna by people.

Based on what archaeological evidence is available, we cannot firmly place people at SWC, or within the FRB, at the time the fossil deposits formed. The oldest archaeological record closest to SWC has an occupation age of ~19 ka at Kenniff Cave[66], Carnarvon Gorge, ~380 km south of SWC. The earliest evidence for human occupation, that is closest to the FRB, comes from two sites over 660 km to the north and northwest of SWC. The two locations, Gledswood Shelter[67] and Ngarrabullgan[61], record human occupation at or younger than 40 ka and occur within the north and westerly-draining Flinders and Mitchell River Basins of Gulf of Carpentaria. Neither basin shares topographic or hydrological connection with the FRB to the southeast.

With no evidence for a human presence within the FRB before ~19 ka, or even more regionally before 40 ka, we cannot implicate people in the extinction of the SWC megafauna. Acknowledging the limited number of archaeological sites older than ~30 ka from eastern Sahul[57,68] we cannot rule out that this absence is a result of significant under sampling, therefore, future concentrated field effort in eastern Sahul may establish an archaeological record older than 40 ka.

## Discussion

Our discoveries provide the first reliably-dated occurrences for several megafauna taxa within the last 50 ka, including all three crocodilians, *Varanus priscus*, *Palorchestes*, *Sedophascolomys*, and at least two macropodids including the largest species known. This increases the total number of megafauna species present at the time of human arrival in Sahul to at least 24. Our new record refutes the claimed extinction timing of *Pallimnarchus*[69] and the modelled extinction timing for *Palorchestes*, *Thylacoleo*, and megafauna representatives of *Macropus*[14]. These new records also document biogeographic extension into north-eastern Australia of several taxa during MIS 3, including *Thylacoleo*, *Phascolonus*, *Diprotodon* and *Protemnodon*. Furthermore, the number of megafauna species identified at our youngest site is greater than that of similar-aged megafauna sites elsewhere in Australia,

increasing the number of taxa surviving the modelled[14] peak of extinction to 13. This amounts to more than half the total number of species now confirmed to be present in Sahul during human occupation.

The age and north-eastern location of our megafauna assemblage is contrary to that which would be expected if continental-wide extinction was human-mediated through direct hunting, where extinctions shortly followed the currently proposed timing and route of human migration across Sahul. Instead, our results favour a scenario of local changes in catchment flow, vegetation structure and fire regime that was not conducive to the survival of aquatic and terrestrial megafauna. These changes should be considered within the context of a regional backdrop of spatio-temporally staggered hydroclimatic deterioration and instability from ~48 ka that occurred across major interior, southern and eastern drainage basins.

Our understanding of the local and regional impacts of environmental deterioration during MIS 3 on megafauna in northern Sahul must also be considered within the context of much longer-term hydroclimate deterioration[70–74] and vegetation change[23,70,75] that was underway since the Middle Pleistocene, well before human occupation. These earlier changes have been implicated in major faunal[4] and floral[23] turnovers in northeast Australia, and yet similar turnovers are not recorded in southern Australian sites spanning similar timeframes[5,76]. Therefore the timing of environmental deterioration and the subsequent responses of fauna and flora differ between regions of Sahul, and have done so across much of the Quaternary. Therefore, more complex scenarios are needed to explain extinctions that take into account these spatio-temporal differences.

## Methods

**Fossil collection and taxonomic identifications**. We employed standard palaeontological and geological procedures during the excavation, including the mapping and recording of positions and associations of fossil remains within the stratigraphic and sedimentological context for each site. Taxonomic identifications were undertaken through indirect comparative assessment via available literature or through direct comparisons with collections held at the Queensland Museum, Museums Victoria, Australian Museum and South Australian Museum (Supplementary Notes 1–4).

**Geochronology**. Dating samples were collected during field excavation and laboratory preparation from 2009 to 2017, with focus on collecting adequate material to undertake multidisciplinary age assessments using four independent dating techniques, specifically optically stimulated luminescence (OSL); radiocarbon ($^{14}$C), uranium series (US), and combined US and Electron Spin Resonance (US-ESR) dating. All sites were sampled; however, due to the poor quality of some samples, not every technique could be applied successfully across all sites. OSL dating has been applied across all sites and via two independent laboratories (Griffith University (GU) and The University of Adelaide (UA)). $^{14}$C and U-series assessments were undertaken by two independent laboratories ($^{14}$C by Beta Analytic (Beta) and Australian National University (ANU)) and U-series by The University of Queensland (UQ) and University of Wollongong (UOW)). ESR assessment was undertaken by a single laboratory at Southern Cross University (SCU).

The matrix depositional age was assessed using OSL, whilst the age of the organic inclusions (teeth/bones/charcoal/organic sediment) was assessed using $^{14}$C, US and ESR. OSL measurements were undertaken independently at GU and UA, and the various burial dose datasets were then analysed using a combined data approach. Associated fossilised teeth were assessed using US, ESR, and combined US-ESR to determine the uptake history of the fossils and allow modelling of finite numerical ages. Attempts were made to radiocarbon date bones, teeth, charcoal and sediment using a variety of pre-treatment methods. Unfortunately, no megafauna could be directly dated due to a lack of collagen. The radiocarbon methods which could be applied to macro-charcoal produced minimum age estimates only. Detailed methodologies are provided in Supplementary Notes 5–8 for all dating techniques used.

**Optically stimulated luminescence**. OSL core samples were collected by QM in 2009, 2011, 2015 and 2016 using extraction techniques and methodologies provided by GU laboratory and through best practice literature (see Supplementary Note 5). This included choosing sampling localities that (1) were freshly exposed

through excavation into cohesive sedimentary matrix; (2) avoided areas showing sediment introduction via recent or old sediment cracking; (3) entirely avoided unconformities, major lithological boundaries and complex sedimentary matrices; (4) were positioned at least 20–30 cm below the exposed ground surface to avoid inaccuracies in gamma dose rates determined using ex situ (laboratory-based) dosimetry techniques. Sediment samples were taken from above and below each core at a radius of between 15 and 20 cm for sedimentological, high-resolution gamma spectrometry (HRGS) and water content assessments. Where possible, OSL core samples were collected in vertical and horizontal sequences to provide both stratigraphic and duplicate samples for any particular layer, including samples taken specifically above, within and below the fossiliferous horizons.

Additional OSL samples were collected by UA in 2017, and included the use of in situ gamma-ray spectrometry (NaI:Tl crystal) to measure external gamma dose rates via the energy windows method[77]. Low-level beta counting was conducted on homogenised sediment collected from the OSL sampling holes to determine external beta dose rates. Conversion and attenuation factors[78–80] were applied to determine the gamma and beta components of the total dose rate. The dose rate contributions from cosmic-rays have been calculated using previous approaches[81]. The beta, gamma and cosmic-ray dose rates have been corrected for long-term sediment moisture contents[82], which were taken to be 40% of the saturated water content following assessments made on samples from freshly-dug pits across various sites.

Quartz grains (212–250-μm diameter) from the 2015 and 2017 samples were purified at UA following standard laboratory procedures[83], including the removal of the alpha-irradiated grain margins through treatment with 48% hydrofluoric acid (40 min)[84]. Equivalent dose ($D_e$) values were determined for single quartz grains using the single aliquot regenerative dose (SAR) procedure[85] and only accepted for dating purposes if they satisfied a series of quality assurance criteria[86]. Single-grain $D_e$ measurements were made using the experimental apparatus described by Dumero et al.[87] and Méndez-Quintas et al.[88] after manually loading quartz grains into standard single-grain aluminium discs drilled with an array of $300 \times 300$-μm holes.

The single-grain quartz $D_e$ measurement procedures employed by GU for the 2009, 2011 and 2016 samples are as described above, with the exception that the 180–212 μm diameter quartz fractions which were isolated for dating using the single-grain quality assurance criteria[89] (see Supplementary Note 5).

We have considered a range of statistical age models for interpreting each single-grain $D_e$ distribution and to derive representative burial dose estimates for the AU and GU OSL samples (central age model (CAM), the three-parameter minimum age model (MAM-3), the four-parameter minimum age model (MAM-4) and the finite mixture model (FMM)[90,91]. The rationale behind the age model choice for each sample is provided in Supplementary Note 5, and is based on consideration of $D_e$ overdispersion, the presence of discrete dose populations, and the statistical suitability of the various age model fits (maximum log likelihood criterion)[92]. The same age model selection criteria have been applied to both the AU and GU $D_e$ datasets to ensure consistency with the final OSL age evaluations. The software, R (luminescent package), was used for the evaluation of the equivalent dose and modelling distributions for CAM and MAM (see Code availability).

**Uranium-series dating**. Several fossil teeth were directly dated using U-series methods (see Supplementary Notes 7 & 8). This included both micro-drilling and laser ablation of dentine with subsequent measurement using multi-collector inductively coupled plasma mass spectrometers (MC-ICP-MS). For micro-drilling, multiple powdered dentine samples were collected with a 1-mm diameter stainless steel hand drill bit through the respective teeth following standard techniques[93]. This allowed for the construction of both U-concentration and $^{230}$Th-age profiles though the teeth, thus, allowing investigation of the mode of U-uptake/loss in the teeth, and essentially, the reliability of the resulting ages. The micro-drilled dentine samples were measured at Radiogenic Isotope Facility, The University of Queensland. using a Nu Plasma MC-ICP-MS following the laboratory's standard protocols[94].

Open-system U-series dating of the teeth via laser ablation MC-ICP-MS was undertaken at the Wollongong Isotope Geochronology Laboratory, University of Wollongong. Laser ablation was performed on a NWR 193 excimer laser coupled to a MC-ICPMS Thermos Plus, at parameters of fluence of 0.9 J/cm$^2$, pulse rate of 10 Hz, spot size of 150 μm and scan speed of 5 μm/s. Helium was used as a carrier gas at a flow rate of 0.9 L/min. Before and after each sample, three rasters were done on NIST612, MK10 (a MIS 7 coral used as primary standard[95]) and MK16 (a MIS 5 coral used as secondary standard). Measured $^{234}$U/$^{238}$U and $^{230}$Th/$^{238}$U isotopic ratios were corrected for elemental fractionation and Faraday cup/SEM yield by comparison with MK10 coral (see above) for which ratios were previously characterised internally by solution analysis. Background subtraction, concentration quantification and ratio corrections were performed using Iolite™ software. Open-system U-Th ages were modelled following the Diffusion-Adsorption-Decay model[96] and using the UThwigl R package[97] (see Code availability).

**Combined U-series and electron spin resonance dating**. ESR dating was performed on a Freiberg MS5000 X-band, coupled to a X-ray Varian irradiator. ESR measurements were performed at 1G modulation amplitude, 2 mW power, 100G

sweep, 100 kHz modulation frequency, while sequence of 90 s, 380 s, 900 s, 1800s, 3600, 7200, 14,400 s was undertaken at 40 kV voltage and 0.5 mA current. For each irradiation step the fragment was measured over 180° in $x$, $y$ and $z$-configurations with a 20° step. Dose response curve (DRC) and equivalent dose ($D_e$) were calculated using the MCDoseE 2.0 program[98]. The external dose was calculated using the values obtained by the gamma spectrometer measurements in the field.

**Radiocarbon dating.** Attempts to date bones, teeth and macro-charcoal at Beta-Analytic using their standard collagen and carbon pre-treatment techniques (Supplementary Note 6) were unsuccessful as no collagen was preserved and macro-charcoal did not survive the processing. Instead bulk organics were extracted and dated from the sediments instead. Radiocarbon dates on sediments are notoriously unreliable given the multiple sources of carbon present, each of which may have a different age. Charcoal found within the fossiliferous layer was dated at the Australian National University using techniques outlined in Supplementary Note 6. Unfortunately, samples were too small and/or too poorly preserved for acid base oxidation-stepped combustion techniques (ABOx-SC), and were cleaned with an acid-base-acid (ABA) protocol. This protocol does not effectively remove contaminants, and the radiocarbon dates on charcoal must be regarded as minimum age estimates. This complicates interpretation as charcoal in fluvial deposits can be considerably older than the deposition of the sediment. We suspect that the macro-charcoal pieces assessed in the fossiliferous layer are reworked within clay-rich rip-up clasts that are observed within the fossiliferous layer but likely derive from older scoured alluvial sediments (Supplementary Notes 4 & 6).

**Palynological assessment.** Clay-rich matrix was sampled from QML1470 SW9 that preserved macroplant fossil remains. These samples were then subsampled for palynological assessment (Supplementary Note 9). Ten subsamples were processed to assess the palynological concentrate to determine the floral diversity of the site (Supplementary Fig. 34 and Supplementary Table 23). Each subsample (~10 g of sediment) was analysed using a method developed for pollen extraction from marine sediments, i.e. low organic content[99] and successfully applied to a range of depositional settings[100]. Samples are deflocculated in sodium pyrophosphate (10%) and then using sodium polytungstate heavy liquid (specific gravity of 1.9) to separate the palynomorph rich organic layer from the mineral clay fraction of the sediment. The pollen concentrate then underwent acetolysis using a 9 to 1 mixture of acetic anhydride and concentrated sulphuric acid. Samples were then placed in glycerol for pollen and micro-charcoal (particles < 125 μm) counting using a ×400 compound light microscope.

**Quality assessment and choice of fossil ages.** Applying selective criteria when choosing or rating the quality of dates for Quaternary-aged fossil fauna is an area of evolving study and refinement[11,12,101,102]. We follow established criteria for assessing the quality of dating techniques used to choose the most reliable ages for megafauna fossils in Sahul[11] dating from MIS 3 (Fig. 1 and Table 1). We apply these same criteria to our own results (Table 2). Those considered to be reliable (e.g. rating of A or A*) were preferred over those that were considered unreliable (e.g. rating of B or C). We then used these same ages to place the youngest records of megafauna within their respective catchments and relate these to available palaeoenvironmental proxies recorded for each catchment and region, thus relating these patterns to our location within the Fitzroy River Basin (Fig. 3).

Youngest high quality-rated megafauna fossil ages are provided in Table 1. Megafauna fossils from Tasmania were directly dated using radiocarbon[103] and were A*-rated[11]. Sediment containing megafauna fossils from southwest Western Australia were dated using associated speleothem (U-series) and sediments (OSL) and were A-rated[30,104]. Egg shell from the LEB has been dated using radiocarbon, OSL and amino acid racemisation, including remains previously identified as the extinct giant flightless bird *Genyornis* and the extant Emu *Dromaius*[105]. The egg shell identified as *Genyornis* has been challenged and is now identified as a species of megapode *Progura*[106]. *Progura* does not meet our mass definition of megafauna; however, we will use it to represent faunal extinction in the LEB. Based on the direct and indirect dating methods applied to these egg shells the ages are A* and A-rated. A specimen of *Zygomaturus* from Willandra Lakes[107] within the MDB was dated using U-series laser ablation applied to bone, returning a minimum age estimate. Sediment found with the specimen was also dated using OSL; however, a lack of field association renders the sediment age uncertain. Therefore a rating of A and B is applied here for the U-series and OSL dates, respectively, with the U-series minimum age preferred. Articulated megafauna remains from the upper Darling River system (Ned's Gully) have been dated using OSL from within the fossil-bearing layer[12]; however, many of the fauna listed from this site[12] lack direct association with the fossil layer[108]. Therefore, we have A-rated the fauna known from the fossil layer and B-rated the fauna with unconfirmed association in the dated fossil layer. Megafauna recovered from Units 6A and 6B, from Cuddie Springs, northern New South Wales, have been dated using A*-rated US-ESR and A-rated OSL methods. The US-ESR ages[69] are consistently older than the OSL ages[12] which may reflect fossil reworking, sediment mixing, or both at the site[69,109,110]. This conclusion has been questioned[53]; however, based on the uncertainty surrounding the OSL ages they are B-rated. We acknowledge the

uncertainty surrounding the age of megafauna at Cuddie Springs and thus present an age range in Table 1, including A* and B-rated ages.

**Reporting summary.** Further information on research design is available in the Nature Research Reporting Summary linked to this article.

## Data availability
The authors declare that data supporting the findings of this study are available in this published article and its supplementary information files. The source data underlying Supplementary Figs. 4 and 9 are provided as a Source Data file.

## Code availability
*UThwigl* R package is available at https://doi.org/10.17605/OSF.IO/D5P7S. R (luminescent package) is available at https://cran.r-project.org/web/packages/Luminescence/index.html.

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

## Acknowledgements

We acknowledge and pay respect to the traditional owners past and present of the lands of the Barada Barna people and thank them for their guidance and contributions. We thank all QM Geosciences staff and volunteers for assistance with this project and in particular thank C. Chiotakis, N. Sands, P. Bishop, E. Cannon, K. Spring, J. Wilkinson, P. Tierney, J. Cramb, N. Newman, I. Mitchell, D. Kennedy, A. Hodgkinson, A. Hollingsworth, T. Clifford, D. Lewis, A. Rozefelds, E. Hatte, C. O'Bryen, Andrey Atuchin, Vlad Konstantinov and Capricorn Caves for their contributions and support. We thank the local land holders of the South Walker and Kemmis Creek areas for their hospitality, in particular the Baulch family. We thank L. Reed, A. Camens, S. Bourne, T. Ziegler, H. Janetzki, P. Couper, A. Amey and C. Janis for their contribution of comparative specimens and imagery. We thank all BHP and BHP Mitsui Coal South Walker Creek Mine staff past and present who have been involved with this project, in particular we thank J. Simpson, B. Clarke, R. Hailstone, J. Schumacher, S. Batten, M. Jones, J. McGill, J. Yanes, A. Cooke, A. Bull, C. Lavey, P. Jeston and V. O'Brien for their support. This project has been supported by the Queensland Museum Foundation, Queensland Museum, BHP, BHP Mitsui Coal and BMA. We thank I. Galloway, S. Miller, J. Thompson, J. Palmer, E. Muller, E. Stolberg J. Hooper, M. Anderson, R. Rew, D. Bunting, M. Partridge, R. Adlard, A. Turley, and S. Baker for this. The OSL dating research undertaken by R.L. and L.J.A. was supported by Australian research Council Future Fellowship grant FT130100195. U-series dating at UQ was supported by Australian Research Council grants DP0881279 and DP120101752 to G.J.P. J.L. is supported by ARC Future Fellowship grant FT160100450. Queensland Xray, St. Vincents Hospital, Princess Alexandra Hospital, Mater Hospital and Museums Victoria are thanked for provision of CT scan data.

## Author contributions

S.A.H., R.A.L., R.L., L.J.A. and T.P. conceived the project, and S.A.H. and R.A.L. supervised the project. All authors contributed to the data interpretation, writing and editing of the paper and Supplementary Information. S.A.H., R.A.L. and J.L. collected the fossil samples. R.L., G.J.P., R.J-B., A.D., S.A.H. and R.A.L. took geochronology samples. S.A.H. and R.A.L. undertook the taxonomic identifications in Supplementary Notes 1–2. S.A.H. and R.A.L. provided the stratigraphic and sedimentological assessment in Supplementary Notes 3–4. R.L., L.J.A., T.P. and J.O. undertook the OSL geochronology analysis in Supplementary Note 5. R.W. undertook the radiocarbon analysis in Supplementary Note 6. G.J.P. undertook uranium-series analysis in Supplementary Note 7. R.J.-B. and A.D. undertook the uranium-series and ESR analysis in Supplementary Note 8. P.M. undertook the palynological assessment in Supplementary Note 9.

## Competing interests

The authors declare no competing interests.
