## [Peer Review File · Nature Communications]

Reviewers' Comments:

Reviewer #1:

Remarks to the Author:

This paper reports results from a truly wonderful new megafauna site in central east Queensland Australia. The study certainly greatly expands our knowledge of megafauna in into tropical Australia. I cannot comment on the taxonomy of the remains, so take these at face value. The site has been dated with multiple techniques and some environmental context is provided by palynology and the identification sporomycete spores in the fluvial sediments that host the remains. I would very much like to recommend acceptance of the paper, but I find it is somewhat confused in the message it is trying to convey, and some of the paleoclimatic interpretations are dubious at best. I think that has arisen by trying to shoehorn the results into the perennial humans versus climate megafaunal extinction debate as the high-profile reason for publication.

The argument presented (in a nutshell) is that megafauna lived beyond the timeframe of 'extinction' suggested elsewhere, and that in the absence of evidence for human presence in the region, the assertion is that climate 'deterioration' must be to blame. So to take these in turn.

(i) Megafauna lived beyond the '42ka' extinction window: first, the central estimate for the extinction window is 42ka, but the error (25th and 75th percentiles) is 37-48 ka. The authors seem to favour, from amongst their ages an estimate of around 40 ka for the bulk of the 'extinction' at their site. I would also note that they have discarded three radiocarbon dates on charcoal (one on a single fragment) all of which have calibrated ages of 42-49ka. These are discarded because charcoal in river sediments might be 'a few thousand years' too old. This is true— a few thousand, but the dates are therefore likely to be older than say 40ka (and these are not shown on the figure in the text). The large number of OSL dates on the sediment (which are reduced to one bar on the figure in the text) are all $>38 \pm 2.5$ ka and up to 66ka, so broadly similar to the radiocarbon, suggesting a minimum age of say ~ 40 ka, or within the bounds proposed for 'extinction' by Saltre et al.. In contrast, the ages least likely to be correct given the open nature of the site are the U-series and maybe ESR dates, which do suggest a younger age (ESR not terribly useful either way), all of which are plotted on the figure in the text. So, the parsimonious interpretation to my mind would be that the small number of U-series and ESR dates are wrong and the large number of OSL (and charcoal) dates places the site at or before 40ka, so not hugely different from the existing suggestion, without placing much credence on the veracity of the claim that any megafauna lingered on much longer at the site. I guess incorporating this into the Signor Lipps modelling of Saltre et al. would presumably reduce the modelled central age a little but not much. The authors say 'Considering the Signor-Lipps effect we conclude that the megafauna at SWC suffered local extinction sometime after 40.5 ± 1.6 ka'. I would say yes, but probably not long after, with no obvious climate driver for extinction at this time.

(ii) The authors claim the absence of evidence for human occupation of the catchment is evidence for a lack of human involvement in extinction. You don't have to go too far northwest of the site to find Gledswood Cave with occupation dated to (I think) 38ka, and given the paucity of sites in general for human occupation, it can be reasonably assumed that humans were present in the catchment.

(iii) The authors claim 'climate deterioration' 'intensified' after 40 ka and point to a number of compilation studies that suggest this (some do, some don't). It is true that the one study within the catchment (Croke et al., 2011), based on OSL dating of river terrace sediments has a gap in dates in the 5,000 year bin centred on 35ka that might indicate drying at this time, but there is a much larger gap in OSL ages between 65 and 75ka, coincident with a stage 4 that has been recently described as 'glacial', with very low sea levels and 'dustier' (De Deckker et al., Quaternary Science Reviews 204 (2019) 187-207), and megafauna did not go extinct at that time a mere 20ka earlier, despite harsher conditions. The evidence from vegetation is equivocal at best and the explanation actually a little

tortured in places. The authors claim (Line 219) 'Vegetation records from ODP820 indicate regional reduction in grasslands (sclerophyll herbs) from ~50 ka reaching 220 their lowest levels at ~35 ka'. Wouldn't low grass pollen in ODP-820 coupled with tree expansion, ordinarily be interpreted as a likely a wetting trend through the interval to 35ka – consistent with the pollen evidence from the site itself, which suggests abundant tree species (eucalypts, casuarina, melaleuca) - and very little grass . This does not suggest a very dry climate at least at the time of sediment accumulation, nor well into the period of 'climate deterioration' that is claimed. Fire will be present at all times given the extensive savannas and the meaning of the charcoal signal in ODP-820 I think is currently difficult to discern, as elsewhere.

So – in summary – a great data set, great site, but I think the interpretation needs a serious rethink. I personally don't care whether extinction was by humans, climate or a combination, but it has to be argued on the evidence, not by tossing out the bits that don't fit.

Other things:

L253 - The authors make the statement in the text: 'Sporormiella is present in low numbers at SW9 (Supplementary Figure 34 and Supplementary Table 24) leaving its relationship with the diverse herbivorous megafauna equivocal'. The data are not on Figure S34, but in the table, SW9 has 59 grains per cm³, and other samples up to 80. These are high levels compared to amounts reported from lake and marine sediments, and I think bolster the argument that Sporormiella is associated with megafauna.

Supplementary table 20 – S-ANU 40224 – units not correct as are given in pMC, not F

Reviewer #2:

Remarks to the Author:

The authors describe the late Pleistocene fossil assemblage of South Walker Creek, northeastern Australia, which preserves at least thirteen extinct and three extant species of megafauna comprising aquatic and terrestrial reptiles and marsupials, including several possible previously un-described species. Dating shows local extinction after 40 ka, coinciding with the onset of climatic and environmental change.

Megafaunal species diversity at 40 ka from South Walker Creek is greater than that of southern regions of similar age, which is contrary to expectations if extinctions followed postulated north to south coastal human migration routes. Therefore, the authors argue the results do not support rapid or synchronous human-mediated continental-wide extinction. Instead they argue the extinctions coincide with regional aridity.

These arguments in this context are original and supported by the data presented. The notion that there is no evidence for humans causing rapid or synchronous extinction of megafauna across the continent is of interest to others in the field, and will promote further debate.

Matt Cupper

Reviewer #3:

Remarks to the Author:

In this manuscript the authors claim to demonstrate that the extinction of megafauna in eastern Sahul

is consistent with environmental change postdating human occupation by a minimum of 10,000 yrs. Overall I think that this argument is well-supported by the evidence they present and there can be no question that this represents a finding of broad interest and significance.

Line 56 and elsewhere: this reviewer is in complete agreement with the proposition that there are "few reliably-dated Upper Pleistocene sites" in Sahul. However, and may be I've missed it, I do think that this a point that requires elaboration. What constitutes a reliably-dated site? On line 172 the authors note that they used direct dating and multiple dating techniques to constrain the age of the fossils they have dated. This seems solid to me, but the definition will likely always be an arbitrary one that in itself has long been hotly debated and see Meltzer & Mead (1985). For example, Roberts et al. 2001 claimed that only sites with articulated remains could be considered as reliably-dated. This determination has been criticized by others, e.g., see Wroe et al. 2004, and has not been widely accepted or used by others. Regardless, I think that the authors should provide a more explicit definition, not only for what they consider to be reliably-dated in the context of their own work, but what they accept as reliably-dated by others.

Line 189: "Considering the Signor-Lipps effect..." I'm really not sure if this effect can be determined on the basis of so few reliably-dated sites, but if it can, how have exactly have they accounted for this?

Lines 243-244: Deterioration in climate and vegetation "would have significantly impacted terrestrial megafauna survival, especially the herbivores and more specifically the grazers". I suggest that anything significantly impacting herbivores must have impacted carnivores that preyed on them. In fact, we would expect the carnivores to go extinct first as herbivore densities fell below densities that could sustain them.

Line 258: "limited evidence of interaction exists", i.e., between humans and extinct megafauna. To the best of my knowledge there is no unambiguous evidence supporting this contention and certainly no kill sites.

Line 280: "contrary to what would expected" I think should be "contrary to that which would be expected", although the editor will be the judge on this.

I think it would be appropriate to have a brief concluding sentence...

Roberts et al. (2001) New Ages for the last Australian megafauna, *Science* 292, 1888

Wroe et al. (2004) Megafaunal extinction in the late Quaternary and the global overkill hypothesis, *Alcheringa* 28, 291

Meltzer & Mead (1985) Dating Late Pleistocene extinctions in "The Cambridge Encyclopedia of Human Evolution, 369-371

Reviewer #4:

Remarks to the Author:

This is an interesting and important paper reporting site/s from Eastern Australia that document a range of megafauna and 'late' survival, while canvassing possible extinction scenarios and timing. I think this is significant for adding to our knowledge of megafauna and their various contexts across Australia (and Sahul).

The paper is well written in places but I find that some of the detail is not clear and some of the

writing is also not clear. The discussion of sites which may have a human association is missing and when mentioned the authors don't appear to be familiar with the data. I think that the authors need to make the whole presentation accessible for a general audience, which at this stage is not.

I note there are no images of the site or material in situ which would have helped get a feel for the location and context.

I have a number of other comments to make and list them in dot point form below.

- Firstly, referring to the 'Upper' Pleistocene is a little confusing. I know name changes have occurred and more generally it has been commonly referred to as the Late Pleistocene c.126ka +, though this paper also includes discussion of Middle Pleistocene as well. (Middle Pleistocene c.781 ka to 126ka). Because of the mix-up with names it is a little confusing. If the authors wish to use the term 'Upper Pleistocene', then I suggest they define it up front (in the title). It just needs clarification.
- Line 48 – the authors must acknowledge that the 40-44kg cut-off for megafauna is a little misleading. There were a range of animals <40-44kg that also went extinct, and were larger than modern day correlates. It was not just confined to larger species.
- Lines 57-58 –I have a real issue with referring to, even if indirectly, an extinction window, initially proposed by Roberts in the 2001 Science paper. This paper is very flawed and has been shown to be statistically invalid. The 7 sites used in the analysis to determine the 'extinction window' were mostly not described in the literature (so where did the samples come from for the dating study, we still don't know). This was pulled apart by Field and colleagues in numerous publications, at least one of those should be cited here as a counter to the Roberts etc assertions, e.g citation 47.
- Line 75 – have these dates been published previously? Isn't this what the paper is about?
- Line 88 – I would prefer to see some description of the site and its setting to be able to understand its context before launching into this description
- Line 89 'Rich deposits of body and trace fossils'???? What does that actually mean?
- Line 90 recovered from where?
- Lines 109-112 I find this a rather 'double negative' convoluted explanation that has no real relevance to this discussion. How can we assess the likely taphonomic factors if we don't even know what the site context is? And why are we talking about species that are not there. Surely the few examples listed above illustrate this point?
- Lines 114-116 – I still don't get why the authors talk about a 'possible new species', it is or it isn't. Perhaps a more conservative description would suffice.
- Line 117 – don't you mean comparison with....postcranial dimensions?
- Line 88-134. I find this section interesting but overly long and convoluted. The one sentence that comprises a paragraph (Lines 125-127) would be much better higher up the section.
- Line 137- this is the first time that four sites are mentioned. This is why we need some context higher up the paper. It assumes knowledge of a whole range of information that hasn't been presented yet.

- Line 140 main channel deposits? Is this a fluvial deposit?? Is there a reason why the authors think this is in situ? What do they actually mean by 'articulated', surely something of this age would have all connective tissue absent, do you mean 'anatomical order'?
- Line 164-171 – Again we are dealing with assumed knowledge with no indication how this relates to context. I find this very frustrating.
- Line 173 – As much as we would like to be able to directly date fossils this is not always possible. U-series and ESR methods as relative dating methods are useful, but only when considered in concert with absolute techniques such as C14, which I note that the authors have tried to do. However, it would be considerably stronger if they could clearly demonstrate the whole picture here. I know I keep reiterating this but the context is as important to assess any assemblage – are we to assume otherwise?
- Line 235 – I would suggest that the detailed work of Mooney et al is not just a 'debate'. It looks at a range of records to support their argument. I also note that the Moss paper cited is an analysis of the ODP820 core, so offshore and conflated. What does the archaeology tell you from this area? I am sure would be hard pressed to place humans here before 40ka on current records, let alone at population levels that would have such effects recorded. To have such a major impact may relate more to changed climatic patterns etc, as stated in the next sentence.
- Line 238 – an interesting observation, if in fact there were enough megafauna around that their loss precipitated such events. We have so little data on this it would be hard to say what population levels actually were, which is clearly important to this scenario.
- Line 257-259 Citation choice a bit odd I would note that the Hamm et al paper has one element of megafauna and not in the cultural levels – what does that mean? It is clear that there is little to no evidence for any interaction of humans and megafauna. The only two sites that may contribute to this are not mentioned here – Nombé Rockshelter and Cuddie Springs. Hardly a comment on a human involvement.
- Line 262-263 – I think you need to mention humans here. Certainly, people don't necessarily 'occupy' that space. A 'human presence' is what you mean, so say that.
- Line 266+ - you have absolutely no evidence of people in this area at this time, the way the authors talk about this implies there is a coexistence. If you look at the fossil record across Sahul there is clearly so little evidence of a human association that it hardly bears commenting on other than to say there is NO evidence of a human association.
- Line 283 – this is selective citation at best. The authors would be better citing the most recent Allen and O'Connell paper, or even going back to the original sources the book called 'Sunda and Sahul' – Birdsell, Horton etc. We really don't know which way people entered Sahul. Looks like multiple migrations and from multiple directions, the north coast of New Guinea being the easiest. With Madjedbebe predating most of the sites on Sahul by a significant period it cannot be relied upon as the only marker. The authors are trying to cover this but need to rethink this a little.
- Line 272+ I would be a little more circumspect about this.
- Line 283-287 – it needs to be considered on a continental scale and in a global context. Extinctions were occurring across Sahul, at different rates and times and it would be best to make that point.

- Line 711 – it must be made clear that this is covering this part of the extinction spectrum. There may have been earlier events – as suggested by the Wroe et al 2013 paper?
- Line 735 – don't you mean Sahul? And if so why haven't you included Nombe. Yes it dates to MIS 2 but surely that is very important. At least in the discussion. Denham and Mountain recently published a new dating study that confirms the original reports and places the megafauna well within the human time-frame. Sutton et al 2010 reviewed fauna from New Guinea.
- Line 743-"the age of units SU6A+B at have been debated due to potential sediment mixing and reworking. We have used US-ESR dates taken directly from fauna to represent the age of these units."

I find this an interesting comment. Perhaps the authors need to read the various papers written by the site investigators (not the ESR dating experts) that clearly describe the deposits, the fauna and their context, and in all cases defend the site against the unsupported assertions of the Roberts/Grun collective. The Grun et al paper may have been the last published on dates, but I note that subsequent papers have challenged their conclusions. Perhaps you should read them? Have you talked to the site investigators? Does the ESR sequence trump C14 and OSL? I don't think so. This is not good scholarship.

- A general comment. This paper has potential, but needs tidying up. More information up front is needed to understand the context before launching into the descriptions. The MNIs reported here are very low, but the impression I get is there was a significant deposits? Also, I would like to see the various age populations reported for the single grain OSL analysis, rather than/or in addition to the weighted average mean. I also note that there is one combined pollen sample because of the few pollen grains surviving in the samples. This should also be explicit. It actually just reports a point in time, especially considering the broad time span and deteriorating conditions that must have prevailed in MIS3. The area covered in this study covers about a third of the Australian continent and probably a couple of environmental zones. It would be useful to understand that. The Sporormiella. What is the take home message from that, other than it is 'there'. As we know that many modern extant fauna dung also harbours these spores, I am not sure what its presence there tell us. Maybe I missed it, but how do the authors see these deposits forming? I don't remember seeing a section on site formation.

Reviewer Remarks in black

Authors Responses in blue

Actions taken within main text and supplementary information in orange

Reviewers' comments:

Reviewer #1 (Remarks to the Author):

We thank all reviewers for their comments and appreciate their respective insight into our work. We feel we have responded to their comments constructively herein and within the main text and Supplementary Information.

This paper reports results from a truly wonderful new megafauna site in central east Queensland Australia. The study certainly greatly expands our knowledge of megafauna in into tropical Australia. I cannot comment on the taxonomy of the remains, so take these at face value. The site has been dated with multiple techniques and some environmental context is provided by palynology and the identification sporomielia spores in the fluvial sediments that host the remains. I would very much like to recommend acceptance of the paper, but I find it is somewhat confused in the message it is trying to convey, and some of the paleoclimatic interpretations are dubious at best. I think that has arisen by trying to shoehorn the results into the perennial humans versus climate megafaunal extinction debate as the high-profile reason for publication.

Our work focuses on introducing a new suite of megafauna from tropical northern Australia, an area with next to no dated megafauna sites from within the period considered vitally important in elucidating the extinction of Sahul megafauna. Our new record securely places a diverse assemblage of megafauna in northern Australia at this crucial time period and provides the first opportunity to test extinction hypotheses. We present our new sites, new fauna and new ages and interpret these results within the available faunal, floral, palaeoclimatic and archaeological data currently available. We place this interpretation within the context of other records from adjacent catchments to the catchment our site belongs, the Fitzroy River Basin (FRB). Given the northern location, the consistency of dating results, and the diverse megafauna found at our sites, this study provides new and valuable insights to better understand the context of megafauna extinction across Sahul.

The argument presented (in a nutshell) is that megafauna lived beyond the timeframe of 'extinction' suggested elsewhere, and that in the absence of evidence for human presence in the region, the assertion is that climate 'deterioration' must be to blame. So to take these in turn.

Our study argues that diverse megafauna, present at our local site in northern Australia, lived beyond the timeframe of the modelled peak of continental-wide extinction for Sahul megafauna. Comparison of our record with others in adjacent catchments suggests a staggered extinction occurring at different times but during a period of sustained environmental deterioration. This environmental deterioration has been recorded across multiple regions using different proxies. At our site, we find no evidence for a human presence, or direct involvement in the extirpation of megafauna. An indirect involvement by humans in megafauna extinction remains equivocal.

(i) Megafauna lived beyond the '42ka' extinction window: first, the central estimate for the extinction window is 42ka, but the error (25th and 75th percentiles) is 37-48 ka. The authors seem to favour, from amongst their ages an estimate of around 40 ka for the bulk of the 'extinction' at their site.

Our data and interpretations do not favour a bulk of extinction occurring at our site around 40 ka. We interpret from our fossil record that megafauna at our location suffered extinction sometime after the youngest reliably-dated fossils (40.1±1.7 ka). This occurs after the proposed/modelled continental-wide 'peak of extinction events' at 42.1 ka (Saltre' et al., 2016). If we ignore the Signor-Lipps effect, and suggest our assemblage was the last of the megafauna in our region, this would be some ~10 kyr younger than the peak of extinction events. (e.g. 50.8 kyr (45.9–57.1 kyr)(Saltre' et al., 2016). Either way, our results do not support the peak in extinctions proposed.

We have updated the age to (40.1+_{-1.7} ka) to clarify the most likely age of our youngest megafauna based on all A* and A-rated dates obtained (n=24). We have altered the tables and text in both the main and supplementary information to reflect this.

I would also note that they have discarded three radiocarbon dates on charcoal (one on a single fragment) all of which have calibrated ages of 42-49ka. These are discarded because charcoal in river sediments might be 'a few thousand years' too old. This is true— a few thousand, but the dates are therefore likely to be older than say 40ka (and these are not shown on the figure in the text). Following criteria set out by Rodríguez-Rey et al (2015), we have classified our dating results to define their reliability, favouring results from techniques that are considered A or A* rating. The charcoal ages are B-rated, at best, whilst the US-ESR and OSL ages are considered A or A* rated. The micro-charcoal that returned the equivocal results are poorly preserved, likely reworked from older sediments, and did not survive the ABOx-SC methods so even though they are calibrated finite ages, they are unreliable estimates for the age of the fossil layer.

We have updated the Methods Section to clarify this.

The large number of OSL dates on the sediment (which are reduced to one bar on the figure in the text) are all $>38 \pm 2.5$ ka and up to 66ka, so broadly similar to the radiocarbon, suggesting a minimum age of say ~ 40 ka, or within the bounds proposed for 'extinction' by Saltre et al..

The OSL ages for the four sites range from 29.7-66 ka with each site defined separately. The youngest of these sites (SW9) would represent the last appearance of the megafauna at SWC and their extinction would be sometime after this. We only illustrate the youngest site age range because this figure compares the reliability-dated ages reported for the youngest megafauna (last appearance) from each region we discuss.

To clarify the difference in the four site ages we have altered Figure 2 (now Figure 3) and the tables to better reflect the age range of each site from SWC.

In contrast, the ages least likely to be correct given the open nature of the site are the U-series and maybe ESR dates, which do suggest a younger age (ESR not terribly useful either way), all of which are plotted on the figure in the text.

So, the parsimonious interpretation to my mind would be that the small number of U-series and ESR dates are wrong and the large number of OSL (and charcoal) dates places the site at or before 40ka, so not hugely different from the existing suggestion, without placing much credence on the veracity of the claim that any megafauna lingered on much longer at the site. I guess incorporating this into the Signor Lipps modelling of Saltre et al. would presumably reduce the modelled central age a little but not much. The authors say 'Considering the Signor-Lipps effect we conclude that the megafauna at SWC suffered local extinction sometime after 40.5 ± 1.6 ka'. I would say yes, but probably not long after, with no obvious climate driver for extinction at this time.

It is clear that there is misunderstanding here in regards to the dating techniques applied and their reliability. Our study represents one of most detailed assessments using U-series and ESR techniques for any single megafauna layer in Australia. As noted above, we plot the A and A* rated ages (US-ESR and OSL weighted mean ages) for the youngest site QML1420 (SW9). The older sites are not plotted on this figure because we are reporting the youngest megafaunal ages for the respective catchments / regions.

US-ESR represents the only dating method (along with radiocarbon dating) that offers a direct age for vertebrate fossils. We applied radiocarbon dating to bones and teeth from our sites; however, due to the lack of collagen, this method was not feasible (See Methods and Supplementary Information). Therefore, at our site, US-ESR represents the most reliable direct-dating technique for megafauna fossils. Combined US-ESR dating provides finite rather than minimum age estimates because the uranium uptake history is explicitly modelled; this circumvents the limitations of open

system behaviour, assuming there has not been any uranium loss. In contrast, the U-series teeth ages on their own represent minimum age estimates, as there are no independent constraints on the dynamics of the post-depositional uranium uptake history. For this reason, we interpret the U-series teeth ages as reliable minimum ages for the faunal accumulation, and the combined US-ESR ages as reliable finite ages for the faunal accumulation. As we have shown in the Supplementary Information, the TIMS results produce consistent U and ^{230}Th age profiles through the teeth, demonstrating that they have not lost U, nor taken any more U up post-depositionally.

In this study we applied US-ESR assessments to five samples from three megafauna taxa primarily to assess whether the fossils were substantially older than the sediments that they were buried in. Although the US-ESR dating uncertainties are large, they are statistically consistent with the OSL results at 1 sigma, and therefore confirm that the fossils are not older than the sediments (i.e. the fossils have not been reworked in the fluvial system). Note that the U-series teeth datasets on their own, yield minimum ages of >24.5 and >25.5 ka, which are in agreement with the finite OSL and US-ESR ages.

The extent of the Signor-Lipps effect is almost impossible to determine from our data without a younger chronology of sites showing a clear decrease in megafauna species diversity over time and a spatially resolved chronological trajectory towards extinction. All we can confidently state is that this diverse megafauna assemblage goes extinct sometime after the last appearance age obtained for our youngest site, which would be sometime after 40.1 ± 1.7 a.

Regarding climate drivers, we would note that we have also examined diverse palaeoenvironmental proxies to show local, regional and continental deterioration that started prior to 40 ka, intensified after 40 ka but, importantly, were sustained over time. Therefore, there is evidence for environmental change of relevance to the megafaunal demise in these basins. These proxies include hydroclimate (monsoon, lake level and river flow activity), vegetation and fire, all of which would have been detrimental to megafaunal survival. We therefore suggest that a staggered and sustained deterioration in environment conditions set in place the key mechanisms that were most detrimental to megafauna survival.

We have defined our methods in greater detail in the Methodology and Supplementary Information. We provide a more holistic age for the youngest site, which combines all dates returned for this site that are considered A* or A-rated for quality (the highest possible). This combined weighted mean age is 40.1 ± 1.7 ka (at 1σ) includes 19 OSL and 5 US-ESR dates. This updated age does not change our overall interpretation.

(ii) The authors claim the absence of evidence for human occupation of the catchment is evidence for a lack of human involvement in extinction.

We do not claim this. Based on the current estimates for human arrival in Australia (ca 65ka- 50ka) and widespread dispersal by 45ka, we state that people may have been present in the landscape at the time that three of our sites were being deposited. However we have no evidence of this occupation at our sites, nor direct evidence of megafauna extirpation by humans.

We have altered the text to better reflect our position on this and integrate this into similar points discussed in the next part of the comment.

You don't have to go too far northwest of the site to find Gledswood Cave with occupation dated to (I think) 38ka, and given the paucity of sites in general for human occupation, it can be reasonably assumed that humans were present in the catchment.

We thank Reviewer 1 for pointing out the very interesting site, Gledswood Shelter. We did know about this locality and the mentioned 38 ka age, however, our literature research only found an unpublished report that was not attainable during this study. We have subsequently found one citation to published dates. Although Gledswood Shelter is the next geographically closest site to SWC that is older than the oldest closest site (Kenniff Cave at 19 ka), it is still situated approximately 660kms to the NW of SWC within in the Flinders Catchment that drains to the northern Gulf of Carpentaria. This catchment does not share a topographic or hydrological relationship with the eastern draining Fitzroy River Basin, where SWC occurs. However, we agree that it does relate to our

broader interpretation of human migration and dispersal and have incorporated it within the main text.

We have incorporated this discussion within the main text including Gledswood Shelter and additionally Ngarrabullgan.

(iii) The authors claim 'climate deterioration' 'intensified' after 40 ka and point to a number of compilation studies that suggest this (some do, some don't).

Respectfully we believe the reviewer has missed the nuance in our argument. We suggest that "the intensification of environmental deterioration starting ~40-41 ka may have therefore been the trigger point for extinction". This statement focuses on our interpretation of environmental factors impacting survival of SWC megafauna within the context of available local and regional palaeoenvironmental proxies of hydroclimate, vegetation change and burning. All of these deteriorated, with sustained and intensified local and regional changes occurring locally after ~40 ka. Across the broader region, there is ample evidence for additional hydroclimatic deterioration occurring within the adjoining catchments of the Lake Eyre and Murray Darling Basins, starting ~50 ka (Figure 2). Therefore, hydroclimatic deterioration is staggered across the three largest catchments in Australia. We consider this to be significant.

It is true that the one study within the catchment (Croke et al., 2011), based on OSL dating of river terrace sediments has a gap in dates in the 5,000 year bin centred on 35ka that might indicate drying at this time, but there is a much larger gap in OSL ages between 65 and 75ka, coincident with a stage 4 that has been recently described as 'glacial', with very low sea levels and 'dustier' (De Deckker et al., Quaternary Science Reviews 204 (2019) 187-207), and megafauna did not go extinct at that time a mere 20ka earlier, despite harsher conditions.

Current understanding of megafauna extinction/s during MIS 4 across Sahul is arguably less well understood than those in MIS 3, so being able to state that megafauna did not go extinct during MIS 4 is unsubstantiated. Most megafauna species remain without reliable chronologies, therefore, we do not know whether their extinctions did or did not occur during MIS 4. We can only state that those megafauna species reliably dated to MIS 3 clearly did not go extinct in MIS 4.

We do not have sites older than 66 ka at SWC, so we cannot and do not speculate on the impacts of MIS 4 climate on the megafauna from this region. The next oldest megafaunal record within the FRB is dated within the middle Pleistocene at Mt. Etna fauna, MIS7-8. The study of (Hocknull et al., 2007) does demonstrate climate-related faunal extinction (including megafauna) and turnover.

We have added in discussion about these older faunal turnovers and their relationship to climate within the main text in regard to the broader temporal context of extinctions within northern Australia.

The evidence from vegetation is equivocal at best and the explanation actually a little tortured in places. The authors claim (Line 219) 'Vegetation records from ODP820 indicate regional reduction in grasslands (sclerophyll herbs) from ~50 ka reaching 220 their lowest levels at ~35 ka'. Wouldn't low grass pollen in ODP-820 coupled with tree expansion, ordinarily be interpreted as a likely a wetting trend through the interval to 35ka – consistent with the pollen evidence from the site itself, which suggests abundant tree species (eucalypts, casuarina, melaleuca) - and very little grass. This does not suggest a very dry climate at least at the time of sediment accumulation, nor well into the period of 'climate deterioration' that is claimed. Fire will be present at all times given the extensive savannas and the meaning of the charcoal signal in ODP-820 I think is currently difficult to discern, as elsewhere.

We recognise the complex relationship between climate-vegetation-megafauna herbivores. The argument regarding the reduction of grasslands and the sustained change to dominantly sclerophyllous forest was not used to demonstrate a drying climate itself as Reviewer 1 suggests. The reduction of grasslands from ~50 ka to ~35 ka was to indicate the overall regional reduction in grasslands and the sustained change to a dominant sclerophyllous forest. This relative low grass and

high sclerophyll forest record is reflected in the palynological assessment of the youngest megafauna site (SW9). We, therefore, suggest the overall regional and local reduction in grasslands as a detrimental component for the diets for the herbivorous megafauna.

We have adjusted the text to more clearly enunciate this observation.

However, Moss et al., (2017) make the point that there is a shift from complex rainforest at 50 ka to drier rainforest at 40 ka in the ODP record – reflecting drier conditions (i.e. reduction in complex rainforest). There is a major peak in charcoal at 38 ka and increased mangrove representation, which is associated with increased soil erosion that may be linked to previous drier conditions.

We have adjusted the text to directly discuss and enunciate this observation of a drying climate.

We agree with the reviewer that the meaning of the charcoal signal at ODP-820 is difficult to discern especially in regards to the overall root cause for increased fire and its relationship to megafauna/vegetation/climate/humans. ODP 820 captures a record relating to the region we are specifically interested in. It shows marked peaks in charcoal, sustained over a ~20,000 year period from ~45-25ka, therefore, regardless of the root cause for increased burning, it is present. We therefore see increased burning as an indicator of environmental deterioration that was initiated in the region prior to 40 ka, and then intensified after 40 ka.

So – in summary – a great data set, great site, but I think the interpretation needs a serious rethink. I personally don't care whether extinction was by humans, climate or a combination, but it has to be argued on the evidence, not by tossing out the bits that don't fit.

Other things:

L253 - The authors make the statement in the text: 'Sporormiella is present in low numbers at SW9 (Supplementary Figure 34 and Supplementary Table 24) leaving its relationship with the diverse herbivorous megafauna equivocal'. The data are not on Figure S34, but in the table, SW9 has 59 grains per cm³, and other samples up to 80. These are high levels compared to amounts reported from lake and marine sediments, and I think bolster the argument that Sporormiella is associated with megafauna.

The variability of concentrations from each sample suggests that interpreting these values and their relationship to megafauna presence or abundance is very difficult, if not impossible, due to the general nature of the fungus being present in fauna that are not megafauna, nor herbivores. It is also necessary to determine the taphonomic settings for the relative preservation of the spores in our fluvial site. Even so, we agree that, with confirmation studies, this might confirm the presence of living megafauna at the time of deposition. This would provide additional support for the primary context of the bone bed and the timing of extinctions. We have adjusted the text to reflect this. However, we do not have results that span a temporal sequence; therefore, we cannot test whether there is a direct link between the dung fungus and megafauna abundance, or megafauna extinction. This is something we will be considering in future work.

We have updated the text to reflect this.

Supplementary table 20 – S-ANU 40224 – units not correct as are given in pMC, not F
The reviewer is incorrect. They are given in F14C. $BP = -5568 * \log(F14C,2)$

Reviewer #2 (Remarks to the Author):

We appreciate the remarks from Reviewer 2 and see that there are no specific issues with this work and they are happy with the work as is.

The authors describe the late Pleistocene fossil assemblage of South Walker Creek, northeastern Australia, which preserves at least thirteen extinct and three extant species of megafauna comprising aquatic and terrestrial reptiles and marsupials, including several possible previously undescribed species. Dating shows local extinction after 40 ka, coinciding with the onset of climatic and environmental change.

Megafaunal species diversity at 40 ka from South Walker Creek is greater than that of southern regions of similar age, which is contrary to expectations if extinctions followed postulated north to south coastal human migration routes. Therefore, the authors argue the results do not support rapid or synchronous human-mediated continental-wide extinction. Instead they argue the extinctions coincide with regional aridity.

These arguments in this context are original and supported by the data presented. The notion that there is no evidence for humans causing rapid or synchronous extinction of megafauna across the continent is of interest to others in the field, and will promote further debate.

Matt Cupper

Reviewer #3 (Remarks to the Author):

We appreciate the remarks from Reviewer 3, and discuss and attend to all of their comments below

In this manuscript the authors claim to demonstrate that the extinction of megafauna in eastern Sahul is consistent with environmental change postdating human occupation by a minimum of 10,000 yrs. Overall I think that this argument is well-supported by the evidence they present and there can be no question that this represents a finding of broad interest and significance.

Line 56 and elsewhere: this reviewer is in complete agreement with the proposition that there are “few reliably-dated Upper Pleistocene sites” in Sahul. However, and may be I’ve missed it, I do think that this a point that requires elaboration. What constitutes a reliably-dated site?

On line 172 the authors note that they used direct dating and multiple dating techniques to constrain the age of the fossils they have dated. This seems solid to me, but the definition will likely always be an arbitrary one that in itself has long been hotly debated and see Meltzer & Mead (1985). For example, Roberts et al. 2001 claimed that only sites with articulated remains could be considered as reliably-dated. This determination has been criticized by others, e.g., see Wroe et al. 2004, and has not been widely accepted or used by others. Regardless, I think that the authors should provide a more explicit definition, not only for what they consider to be reliably-dated in the context of their own work, but what they accept as reliably-dated by others.

We have followed the quality rating criteria set out by Rodríguez-Rey et al (2015), who determined the reliability of different dating techniques in relation to methodological considerations and the stratigraphic association of the fossil material being assessed. We could not locate Meltzer and Mead (1985) or a citation.

We have updated the text in the Methodology to better describe these criteria and how we applied them.

Line 189: “Considering the Signor-Lipps effect...” I’m really not sure if this effect can be determined on the basis of so few reliably-dated sites, but if it can, how have exactly have they accounted for this?

The Reviewer is correct in that we cannot determine the extent of the Signor-Lipps effect in a statistical sense for the basin due to the low number of sites. For this reason, we only consider the

overarching implication of the Signor-Lipps effect when interpreting the extinction timing for the local megafauna. We cannot claim to have found the fossilised remains of the very last megafauna individuals from this environment. The last appearance ages obtained in this study do not equate to true extinction ages; hence we consider our results to provide maximum age estimates for the true extinction 'event'. In other words, we can reasonably state that extinction occurred sometime after the youngest reliably dated megafauna remains. This interpretation remains conceptually valid irrespective of whether we can or cannot quantify the Signor-Lipps effect.

We have altered the text to better describe this point.

Lines 243-244: Deterioration in climate and vegetation "would have significantly impacted terrestrial megafauna survival, especially the herbivores and more specifically the grazers". I suggest that anything significantly impacting herbivores must have impacted carnivores that preyed on them. In fact, we would expect the carnivores to go extinct first as herbivore densities fell below densities that could sustain them.

We agree with the reviewer on this point. The carnivorous taxa remain diverse at our youngest site, with the only possible loss of "*Quinkana*", a large, possibly terrestrial crocodylian. The diversity of carnivores, which are almost entirely reptilian, reflect a unique ecological makeup for the megafauna. Therefore, the prey requirements / carrying capacity are likely to be different to those from mammalian carnivore dominated faunas, and the signal of carnivore loss may not be as evident in these reptile-dominated megafauna assemblages.

We have altered the text to better reflect these points.

Line 258: "limited evidence of interaction exists", i.e., between humans and extinct megafauna. To the best of my knowledge there is no unambiguous evidence supporting this contention and certainly no kill sites.

Yes, we agree with this statement that there are no kill sites; however, there are sites that possess archaeology and megafauna remains, all of which are equivocal but present.

We have altered the text to reflect this point.

Line 280: "contrary to what would expected" I think should be "contrary to that which would be expected", although the editor will be the judge on this.

We have changed the text.

I think it would be appropriate to have a brief concluding sentence...

We have added a brief concluding sentence incorporating other remarks from other reviewers also.

Roberts et al. (2001) New Ages for the last Australian megafauna, *Science* 292, 1888

Wroe et al. (2004) Megafaunal extinction in the late Quaternary and the global overkill hypothesis, *Alcheringa* 28, 291

Meltzer & Mead (1985) Dating Late Pleistocene extinctions in "The Cambridge Encyclopedia of Human Evolution, 369-371

Reviewer #4 (Remarks to the Author):

We thank all reviewers for their comments and appreciate their respective insight into our work. We feel we have responded to their comments constructively herein and within the main text and Supplementary Information.

This is an interesting and important paper reporting site/s from Eastern Australia that document a range of megafauna and 'late' survival, while canvassing possible extinction scenarios and timing. I

think this is significant for adding to our knowledge of megafauna and their various contexts across Australia (and Sahul).

The paper is well written in places but I find that some of the detail is not clear and some of the writing is also not clear. The discussion of sites which may have a human association is missing and when mentioned the authors don't appear to be familiar with the data.

We appreciate that some readers would be specifically interested in those sites that may have evidence for a human association; however, we are not offering a systematic review of the archaeological record in relationship to megafauna fossils. Instead, the focus of this work is on introducing new tropical northern Australia megafauna that have been dated using a multi-disciplinary approach. This allows us to place our megafaunal assemblage in a chronological and geographical context, and to infer possible extinction mechanisms for these fauna. We then look more broadly at adjacent catchments to see whether similar mechanisms may be replicated. What we find is that environment deterioration occurs in similar ways but is chronologically and geographically staggered.

I think that the authors need to make the whole presentation accessible for a general audience, which at this stage is not.

Within the constraints of word count and staying on topic, our presentation focusses on drawing upon a very large dataset, as demonstrated by the >100 page Supplementary Information. We have summarised this data and present the main arguments in the body of the text, and provide extensive details supporting these arguments in the Supplementary Information.

I note there are no images of the site or material in situ which would have helped get a feel for the location and context.

Within the constraints of figure size, number and data presentation for the main body, we felt that this did not add to the key foci of this work. We provide detailed site maps, specimen locations, taphonomic details and locations of dating samples in the SI.

We have added a new Figure (now Figure 2) and text to the main text including site location, imagery and images of fossil remains in context in addition to the detailed context in the Supplementary Information.

I have a number of other comments to make and list them in dot point form below.

- Firstly, referring to the 'Upper' Pleistocene is a little confusing. I know name changes have occurred and more generally it has been commonly referred to as the Late Pleistocene c.126ka +, though this paper also includes discussion of Middle Pleistocene as well. (Middle Pleistocene c.781 ka to 126ka). Because of the mix-up with names it is a little confusing. If the authors wish to use the term 'Upper Pleistocene', then I suggest they define it up front (in the title). It just needs clarification.

We have provided the age range in the main text for the Upper Pleistocene as defined by the International Commission on Stratigraphy in their 2019 International Chronostratigraphic Chart.

- Line 48 – the authors must acknowledge that the 40-44kg cut-off for megafauna is a little misleading. There were a range of animals <40-44kg that also went extinct, and were larger than modern day correlates. It was not just confined to larger species.

We agree with the reviewer about this point.

We have edited the main text to include our definition of megafauna.

- Lines 57-58 –I have a real issue with referring to, even if indirectly, an extinction window, initially proposed by Roberts in the 2001 Science paper. This paper is very flawed and has been shown to be statistically invalid. The 7 sites used in the analysis to determine the 'extinction window' were mostly

not described in the literature (so where did the samples come from for the dating study, we still don't know). This was pulled apart by Field and colleagues in numerous publications, at least one of those should be cited here as a counter to the Roberts etc assertions, e.g citation 47.

Although we do not specifically refer to an extinction window, our intention in this sentence is to draw the reader's attention to the period of time (MIS3) that is considered by most to include the final extinction of the Upper Pleistocene megafauna taxa. The 42 ka peak extinction referred to in this sentence is not based on the (Roberts et al., 2001) dataset but rather the larger and more recent chronological synthesis study of (Saltré et al., 2016), which incorporates dating quality rating criteria and accounts for the Signor-Lipps effect.

We have added in additional citations representing other works that discuss extinction timing.

- Line 75 – have these dates been published previously? Isn't this what the paper is about?

The dates have not been published previously and this is what the paper is about. We are unclear as to what this comment pertains. We assume it is in reference to the use of 'Upper', in Upper Pleistocene and that the dates are what have confirmed these taxa as Upper and not from Middle or Early Pleistocene?

We have removed Upper from Upper Pleistocene in this section.

- Line 88 – I would prefer to see some description of the site and its setting to be able to understand its context before launching into this description

We provide detailed geological, depositional and site descriptions in the Supplementary Information. As above, we have added a new Figure (now Figure 2) and text to the main text including site location, imagery and images of fossil remains in context in addition to the detailed context already available in the Supplementary Information.

- Line 89 'Rich deposits of body and trace fossils'???? What does that actually mean?

Changed rich to fossiliferous

- Line 90 recovered from where?

Changed text to "recovered from SWC"

- Lines 109-112 I find this a rather 'double negative' convoluted explanation that has no real relevance to this discussion. How can we assess the likely taphonomic factors if we don't even know what the site context is? And why are we talking about species that are not there. Surely the few examples listed above illustrate this point?

We altered the text to better clarify our statements.

- Lines 114-116 – I still don't get why the authors talk about a 'possible new species', it is or it isn't. Perhaps a more conservative description would suffice.

All species concepts are arbitrary constructs defined by taxonomists to determine phenotypic or genotypic diversity. They are therefore only ever hypotheses and as such are only ever possibilities. In the context of this work, however, we are demonstrating a level of taxonomic uncertainty regarding the possibility of morphologically distinct taxa. These possibilities are discussed further in the Supplementary Information.

- Line 117 – don't you mean comparison with....postcranial dimensions?

We altered the text to better clarify our statement.

- Line 88-134. I find this section interesting but overly long and convoluted. The one sentence that comprises a paragraph (Lines 125-127) would be much better higher up the section.

We altered the text to better clarify our statement.

- Line 137- this is the first time that four sites are mentioned. This is why we need some context

higher up the paper. It assumes knowledge of a whole range of information that hasn't been presented yet.

As above, we have added a new Figure (now Figure 2) and text to the main text including site location, imagery and images of fossil remains in context in addition to the detailed context already available in the Supplementary Information.

- Line 140 main channel deposits? Is this a fluvial deposit?? Is there a reason why the authors think this is in situ? What do they actually mean by 'articulated', surely something of this age would have all connective tissue absent, do you mean 'anatomical order'?

We describe the depositional settings and taphonomic setting throughout the main text and within the Supplementary Information. The articulated elements remain in articulation to one another demonstrating lack of movement of elements away from one another. `

We have altered the text to better clarify this and provided examples within the new Figure 2.

- Line 164-171 – Again we are dealing with assumed knowledge with no indication how this relates to context. I find this very frustrating.

The detailed contexts for these statements are within the Supplementary Information.

We have provided further text to clarify this.

- Line 173 – As much as we would like to be able to directly date fossils this is not always possible. U-series and ESR methods as relative dating methods are useful, but only when considered in concert with absolute techniques such as C14, which I note that the authors have tried to do. However, it would be considerably stronger if they could clearly demonstrate the whole picture here. I know I keep reiterating this but the context is as important to assess any assemblage – are we to assume otherwise?

We do not consider these comments here to be substantiated. Both U-series and ESR dating techniques are numerical ('absolute') dating methods (not relative dating methods), and similar to radiocarbon dating (also an absolute dating technique), all three methods have advantages and limitations. For this reason we have adopted a multi-method dating approach in our study, utilising all currently available and accepted radiometric dating methods. This has allowed us to generate several independent ages estimate for the site, which gives insight not only into the age of the fossil deposit but also to the potential primary context of fossils within complex depositional settings. There is an extensive literature that demonstrates both U-series and ESR techniques have been successfully applied to teeth and bones. Moreover, the underlying physical mechanisms have been extensively studied and we present all required data to reinterpret the measurements should new studies reveal problems with the current foundation of the method.

- Line 235 – I would suggest that the detailed work of Mooney et al is not just a 'debate'. It looks at a range of records to support their argument.

We have altered the text to 'challenged'.

I also note that the Moss paper cited is an analysis of the ODP820 core, so offshore and conflated. An offshore record provides a better regional signal (avoiding conflation) and thus a better representation of the regional palaeoenvironmental signal versus terrestrial signals. This has been demonstrated for ODP 820 by Moss et al., (2005). An additional advantage is that it possesses independent age control provided by marine oxygen isotope record, thereby avoiding problems with radiocarbon.

What does the archaeology tell you from this area? I am sure would be hard pressed to place humans here before 40ka on current records, let alone at population levels that would have such effects recorded. To have such a major impact may relate more to changed climatic patterns etc, as stated in the next sentence.

We discuss archaeological context this further down in the text.

- Line 238 – an interesting observation, if in fact there were enough megafauna around that their loss precipitated such events. We have so little data on this it would be hard to say what population levels actually were, which is clearly important to this scenario.

We agree that we do not currently know the relationship between megafauna populations / grazing pressure and frequency of fire. It is relevant to suggest another scenario that may explain intensified burning throughout this period.

- Line 257-259 Citation choice a bit odd I would note that the Hamm et al paper has one element of megafauna and not in the cultural levels – what does that mean? It is clear that there is little to no evidence for any interaction of humans and megafauna. The only two sites that may contribute to this are not mentioned here – Nombe Rockshelter and Cuddie Springs. Hardly a comment on a human involvement.

We have updated the citation list to include additional references to Cuddie Spring, however, to our knowledge the updated context of humans and megafauna at Nombe Rockshelter have not been published in mainstream, peer-reviewed scientific journals and therefore we cannot offer comment on this, it remaining equivocal at this stage.

- Line 262-263 – I think you need to mention humans here. Certainly, people don't necessarily 'occupy' that space. A 'human presence' is what you mean, so say that.

We altered the text to better clarify our statement.

- Line 266+ - you have absolutely no evidence of people in this area at this time, the way the authors talk about this implies there is a coexistence. If you look at the fossil record across Sahul there is clearly so little evidence of a human association that it hardly bears commenting on other than to say there is NO evidence of a human association.

We agree with the reviewer that there is very little evidence of a human association with megafauna in Sahul, however, we cannot say that there is no association. The relationship between humans and megafauna is very contentious. In comparison to a separate reviewer, their opinion is very much different in that there is a clear YES to an association and that humans are clearly involved. It is our view that discussion of possible human presence at our sites is warranted because it is expected. We, do however, remain exceptionally broad in our assumptions, and conclude that if humans were present they left no archaeological signature in the fossil deposits.

We have attended to this discussion of potential human-megafauna associations along with other reviewer's remarks. We believe keeping this discussion in the work is relevant based on the expectation that such a discussion would be needed regardless of how speculative it may be. We have been more explicit regarding our observations at SWC vs the broader context of human occupation, migration and association with megafauna.

- Line 283 – this is selective citation at best. The authors would be better citing the most recent Allen and O'Connell paper, or even going back to the original sources the book called 'Sunda and Sahul' – Birdsell, Horton etc. We really don't know which way people entered Sahul. Looks like multiple migrations and from multiple directions, the north coast of New Guinea being the easiest. With Madjedbebe predating most of the sites on Sahul by a significant period it cannot be relied upon as the only marker. The authors are trying to cover this but need to rethink this a little.

We have identified a wrong citation here and fixed this. The number and selection of citations is heavily restricted for these types of manuscripts so we used the two citations that are most recent and present modelled pathways across the continent.

It is clear from all models that the migration of people has occurred from the north and possibly in a variety of directions and possibly in single or multiple migrations. Coming from the north, it would be reasonably expected then that extinction chronology would follow this route, if extinction directly

involved human extirpation. Our site, in northern Australia and at 40 ka, demonstrates that this is clearly not the case.

We have provided additional citations and discussions of the most recent research and discussions surrounding proposed migration routes for humans across Sahul.

- Line 272+ I would be a little more circumspect about this.

We are unclear as to why we would need to be circumspect about these statements, they are supported fully by our study.

- Line 283-287 – it needs to be considered on a continental scale and in a global context. Extinctions were occurring across Sahul, at different rates and times and it would be best to make that point. We agree that extinctions were occurring across Sahul and very likely at different rates and different times, although no single site records just what these rates were or their ultimate timing. Therefore, we need to focus on what we can state based on our new study, and placing this in context of the region it represents. This is what we do in relation to developing a testable scenario for a spatio-temporally staggered deterioration in environment across the vast inland and eastern regions of Australia. Our new data cannot extend to areas not directly related to it.

- Line 711 – it must be made clear that this is covering this part of the extinction spectrum. There may have been earlier events – as suggested by the Wroe et al 2013 paper?

We do see extinctions across the Middle to Upper Pleistocene in Australia, and we see significant extinctions occurring prior to the Upper Pleistocene with the FRB. Indeed we see the greatest extinctions of the Middle Pleistocene in this region as reported by Hocknull et al 2007. However, there is a significant gap in the faunal record between these sites and those of the 40 ka. Therefore, their relevance will only be useful once we have a record that includes the intervening 200 ka between the extinction of these Middle Pleistocene taxa and those of the Upper Pleistocene. Therefore, in the context of this particular work, we do not see a firm relevance to discussing these previous extinctions. However, for completeness and to address the differences in Middle Pleistocene extinctions across Sahul we have added in a discussion of these previous extinctions within the region we are focused.

We have updated the main text to reflect this discussion point.

- Line 735 – don't you mean Sahul? And if so why haven't you included Nombe. Yes it dates to MIS 2 but surely that is very important. At least in the discussion. Denham and Mountain recently published a new dating study that confirms the original reports and places the megafauna well within the human time-frame. Sutton et al 2010 reviewed fauna from New Guinea.

We considered the excellent work being undertaken on the Nombe Rockshelter, in particular the reviewed dating, however, due to the ambiguity of the ages of the megafauna layers not yet published we did not and feel that we cannot include it.

- Line 743-"the age of units SU6A+B at have been debated due to potential sediment mixing and reworking. We have used US-ESR dates taken directly from fauna to represent the age of these units."

I find this an interesting comment. Perhaps the authors need to read the various papers written by the site investigators (not the ESR dating experts) that clearly describe the deposits, the fauna and their context, and in all cases defend the site against the unsupported assertions of the Roberts/Grun collective.

We have read all papers pertinent to the dating and depositional context of Cuddie Springs, along with its fauna. For the verification of faunal identifications, we relied on the available literature for species from this site. The available literature leaves uncertainty around these taxonomic identifications, with the exception of *Genyornis* and *Diprotodon* at Cuddie Springs. We considered

removing this site from this present work due to this uncertainty, however, we understand the significance of it within the context of the current understanding of the timing and context of megafauna extinction in Australia, so retained it.

Similarly, there is uncertainty surrounding the reliability of dating results for Cuddie Springs that do not provide the level of detail needed for us to fully assess their reliability. Therefore, without this published detail, the decision of reliability and species lists need to use established criteria, such as those outlined by Rodríguez-Rey et al (2015). We have applied these same criteria to all of the sites we discuss, including our own data, presenting taxonomic identification and dating results in detail (see Supplementary Information). We concluded for Cuddie Springs that the US-ESR results for units SU6A+B represent the most reliable date for the megafauna from these layers.

We have updated Figure 2 (now Figure 3) to better reflect this uncertainty of ages. However, the overall spread of dates does not alter our interpretation that extinctions within the Murray-Darling Basin occur sometime at the onset or during hydroclimatic deterioration and instability.

The Grün et al paper may have been the last published on dates, but I note that subsequent papers have challenged their conclusions. Perhaps you should read them?

We are unaware of subsequent research that directly challenges the validity of the results or presents new ones.

Have you talked to the site investigators?

To comment in a paper, the information/opinions need to be published. We, therefore, relied on published records for our study.

Does the ESR sequence trump C14 and OSL? I don't think so. This is not good scholarship.

Our manuscript does not specifically review Cuddie Springs or provide a detailed discussion of the dating uncertainties of that site, simply because this is well outside the scope of our study. However, applying the same methodology and criteria we have applied to our own results, in regards to using the most reliable-ages (Rodríguez-Rey et al., 2015), we consider the direct-dating of the megafauna as most reliable. The C14 dated using charcoal, which is a B-rated technique and the OSL dates are likely affected by sediment mixing this uncertainty gives it a B-rating. Therefore we use the published US-ESR results of (Grün et al., 2010), which are the most recently published ages for this site.

We have updated Figure 2 (now Figure 3) to reflect this uncertainty.

- A general comment. This paper has potential, but needs tidying up. More information up front is needed to understand the context before launching into the descriptions.

We believe that that while the contextual site information are critical for generating the data and interpretations of our study, they are not necessary to understanding the broad arguments we make in the body of the paper, and would be of limited interest to the general reader. Thus, we list them only in the SI.

As above, we have added a new Figure (now Figure 2) and text to the main text including site location, in situ site/specimen imagery and images of fossil remains in context supporting the variety of preservation and taphonomic points covered in the main text. This is in addition to the detailed context already available in the Supplementary Information.

The MNIs reported here are very low, but the impression I get is there was a significant deposits?

Yes, the MNIs are low, and this is because most of the remains are associated from single individuals. By themselves, low MNIs do not reflect the significance of a deposit. The sites contain abundant remains of diverse new fauna, securely dated and from a region of Australia without a fossil record, that reflects its significance. We also show in the main text that modelled predictors of

discoverability would suggest that these sites should not exist or are exceptionally rare, therefore, to find them and preserving a diverse fossil record is highly unexpected.

Also, I would like to see the various age populations reported for the single grain OSL analysis, rather than/or in addition to the weighted average mean.

All of the data pertinent to the interpretation of the OSL datasets have already been provided in the SI. To clarify, the age model selection procedure for the single-grain D_e datasets is sample-specific and founded on objective statistical criterion, namely the maximum log likelihood ratio test of Arnold et al. (2019). The results of this age model selection procedure are detailed in Table S12. For all but two of the thirty four OSL samples, this age model selection criterion unequivocally supports use of the central age model (weighted mean) for D_e determination. Presentation of alternative age models that are not considered valid in this particular depositional context or are not supported by the datasets on statistical grounds would not offer any advantage in this study. On the contrary, this information could add confusion for the non-specialist reader. Only two of the OSL samples presented in this study contained discrete D_e populations that required consideration of an alternative age model (finite mixture model). These samples have been discounted from the final analyses on methodological grounds, as detailed in the SI. Nevertheless, Tables S13 and S14 provide full details of the finite mixture model D_e values obtained for these two samples, and the D_e populations observed for these samples are also provided in Figures S24 and S26.

The requested information is either already provided in the SI or is not relevant from an OSL dating methodological or interpretative perspective.

I also note that there is one combined pollen sample because of the few pollen grains surviving in the samples. This should also be explicit.

This is provided within the Supplementary Information.

It actually just reports a point in time, especially considering the broad time span and deteriorating conditions that must have prevailed in MIS3. The area covered in this study covers about a third of the Australian continent and probably a couple of environmental zones. It would be useful to understand that.

We utilise the pollen study at SWC locally to help our interpretation of the local environment at our youngest site, demonstrating the relative proportions of species. Therefore, the area of our study we use the pollen from SWC to interpret is the local environment. We then compare this to ODP820, the most appropriate regional comparison.

The *Sporormiella*. What is the take home message from that, other than it is 'there'. As we know that many modern extant fauna dung also harbours these spores, I am not sure what its presence there tell us. Maybe I missed it, but how do the authors see these deposits forming? I don't remember seeing a section on site formation.

We provide detailed descriptions of the site depositional context and the *Sporormiella* and its context in the Supplementary information.

Grün, R., Eggins, S., Aubert, M., Spooner, N., Pike, A.W.G., Müller, W., 2010. ESR and U-series analyses of faunal material from Cuddie Springs, NSW, Australia: implications for the timing of the extinction of the Australian megafauna. *Quaternary Science Reviews* 29, 596-610.

Hocknull, S.A., Zhao, J.-X., Feng, Y.-X., Webb, G.E., 2007. Responses of Quaternary rainforest vertebrates to climate change in Australia. *Earth and Planetary Science Letters* 264, 317-331.

Roberts, R.G., Flannery, T.F., Ayliffe, L.K., Yoshida, H., Olley, J.M., Prideaux, G.J., Laslett, G.M., Baynes, A., Smith, M.A., Jones, R., Smith, B.L., 2001. New ages for the last Australian megafauna: continent-wide extinction about 46,000 years ago. *Science* 292, 1888-1892.

Rodríguez-Rey, M., Herrando-Pérez, S., Gillespie, R., Jacobs, Z., Saltré, F., Brook, B.W., Prideaux, G.J., Roberts, R.G., Cooper, A., Alroy, J., Miller, G.H., Bird, M.I., Johnson, C.N., Beeton, N., Turney, C.S.M.,

Bradshaw, C.J.A., 2015. Criteria for assessing the quality of Middle Pleistocene to Holocene vertebrate fossil ages. *Quaternary Geochronology* 30, 69-79.

Saltré, F.d.r., Rodríguez-Rey, M., Brook, B.W., Johnson, C.N., Turney, C.S.M., Alroy, J., Cooper, A., Beeton, N., Bird, M.I., Fordham, D.A., Gillespie, R., Herrando-Pérez, S., Jacobs, Z., Miller, G.H., Nogue's-Bravo, D., Prideaux, G.J., Roberts, R.G., Bradshaw, C.J.A., 2016. Climate change not to blame for late Quaternary megafauna extinctions in Australia. *Nature communications* 7, 10511.

Reviewers' Comments:

Reviewer #1:

Remarks to the Author:

This is a revised version of a paper I reviewed previously and the authors have modified some of the text to reflect some of the concerns I had with the earlier version, concerns shared with one of the other reviewers in several cases.

The authors have provided a rebuttal of the concerns I had with the original, and I remain unconvinced by some responses, as in some cases have simply re-iterated their original position in different words. The response is long, so I will simply note a couple of things below in this regard.

At the end of the day though, this is a very nice and thorough piece of work in most respects, and I'm not sure my opinion should trump theirs. I am therefore happy to recommend publication of this version and let history be the judge, the data is fine, it's the nuance of the interpretation that in some parts I take issue with. I note that the authors now claim a minimum reliable age of 40.1 ± 1.7 ka, which is about what I would have thought is a reasonable firm claim anyway. I'd also add that I like the figures and tables as currently configured in the revised version, even if they might be pushing the page limit.

Specifically, on two points of several I could draw out as illustration:

(i) I pointed out that there were earlier MIS4 periods when climate deterioration, in the study area, using the same study as the authors, could have rendered the megafauna extinct but didn't. The authors state: 'being able to state that megafauna did not go extinct during MIS 4 is unsubstantiated'. This does not address the issue I raised – clearly megafauna did not go extinct during MIS4, because the authors present data that they continued to exist in MIS-3. My point was that climate deterioration as a sole driver of extinction in MIS-3 or any other time after human arrival, is probably not a viable explanation.

(ii) I made the point that it was not appropriate to discount the radiocarbon dates on charcoal, which might be a few millennia older than the sediment, but not more. The authors responded that the charcoal dates were discounted because they have applied the quality rating scale of Rodrigues-Rey et al (2015), and the dates are 'b' quality and they only used a and a*. This is technically correct, but my point was how much older could the charcoal dates reasonably be than the age of the sediment (and the fossils), and the answer is a few millennia maximum and probably less. The authors then say the charcoal was 'badly preserved', and again, fine, but that would make the ages more likely minimum ages, and the dates do provide a constraint that suggest probably not younger than 40 and possibly quite a bit older. The 'reworked' argument I think is weak, though not entirely discountable I guess.

(iii) I also note, on reflection, that the Gledswood dates I referred to are unpublished I think, so no reason for the authors to have known about them. I do think its not credible to claim that people weren't around in the catchment – my view is that people were everywhere by then, but again that's only my opinion.

Reviewer #3:

Remarks to the Author:

I am happy with the authors' responses and changes to the MS. I think it's good to go.

Reviewer #4:

Remarks to the Author:

I have read both the revised manuscript and the SI. The increased clarification by changes in the text is very helpful. With the excuse that the authors could not find the Mead and Metzler reference I would advise they widen their search and perhaps refer to the other Mead and Meltzer reference from the same year. A quick google search will do it and then a visit to the library.

I am still critical of the treatment of the only two sites where megafauna and a record of humans co-occurring in Sahul have been identified: Nombe Rockshelter and Cuddie Springs. With Nombe, the recent dating project has confirmed the ages of the Unit where the megafauna were found. It is interesting that the human record is found with fragmented faunal remains. Nonetheless when an interaction between humans and megafauna cannot be demonstrated, it does indicate that megafauna survived in the highlands till quite late. There is no excuse for ignoring these results. The evidence is clear.

Not everyone agrees that the Salitre et al dating confidence attributions are an accurate representation of the dating confidence. To dismiss the Cuddie Springs evidence as possibly disturbed is a clear indication that the authors have either not read the relevant papers or not understood them.

Furthermore, it is somewhat annoying to give ESR/U-series methods such an elevated position in the dating suite. How this has happened I am not sure, yet in other studies ESR-U series has been up to 10,000 years older than other dating (OSL/C14) methods. Are the authors now suggesting via Salitre et al that the Mungo deposits are 20,000 years older than the accepted OSL and C14, or that the Devils Lair sequence is also much older? Do they not see the consequences of promoting such interpretations because it fits their thesis? This is a rhetorical question of course.

I would also suggest that the authors be more circumspect about the presence of humans. Until it is demonstrated (by empirical evidence) to be so then you cannot claim they were probably there. The work of Alan Williams has shown that the population levels across Sahul at this time were very low. Does that not indicate something? Furthermore, the fact there are only two megafauna sites in Sahul with an associated human record, which by the way these authors dismiss, does that also tell you that either megafauna or humans were pretty thin on the ground?

Reviewer Comments – Responses

We thank the reviewers for their additional comments and provide responses to these below.

REVIEWERS' COMMENTS:

Reviewer #1 (Remarks to the Author):

We thank Reviewer #1 for their additional comments and we are glad that our new version addresses many of their issues. We appreciate their acknowledgment that our work is thorough.

This is a revised version of a paper I reviewed previously and the authors have modified some of the text to reflect some of the concerns I had with the earlier version, concerns shared with one of the other reviewers in several cases.

The authors have provided a rebuttal of the concerns I had with the original, and I remain unconvinced by some responses, as in some cases have simply re-iterated their original position in different words. The response is long, so I will simply note a couple of things below in this regard.

At the end of the day though, this is a very nice and thorough piece of work in most respects, and I'm not sure my opinion should trump theirs. I am therefore happy to recommend publication of this version and let history be the judge, the data is fine, it's the nuance of the interpretation that in some parts I take issue with. I note that the authors now claim a minimum reliable age of 40.1 ± 1.7 ka, which is about what I would have thought is a reasonable firm claim anyway. I'd also add that I like the figures and tables as currently configured in the revised version, even if they might be pushing the page limit.

Specifically, on two points of several I could draw out as illustration:

(i) I pointed out that there were earlier MIS4 periods when climate deterioration, in the study area, using the same study as the authors, could have rendered the megafauna extinct but didn't. The authors state: 'being able to state that megafauna did not go extinct during MIS 4 is unsubstantiated'. This does not address the issue I raised – clearly megafauna did not go extinct during MIS4, because the authors present data that they continued to exist in MIS-3. My point was that climate deterioration as a sole driver of extinction in MIS-3 or any other time after human arrival, is probably not a viable explanation.

We believe we have addressed this issue in the discussion. We reference major climatic changes occurring during the Middle Pleistocene that are implicated for major faunal turnovers in northern Sahul that included megafauna. This scenario differs to faunal responses during the Middle and Late Pleistocene of southern Australian regions; therefore, we need to consider the extinction of fauna relative to the regional / local available proxies and guide our interpretations based on this. We, therefore, cannot directly test this at South Walker Creek because we do not yet have sites that are this old. However, sites within the Fitzroy River Basin do demonstrate local faunal and megafaunal extinction during the Middle Pleistocene, occurring over periods of past climatic change.

We have evidence for dramatic environmental change occurring at, or during, the period of megafauna extinctions in our region, occurring both during the Middle and Late Pleistocene. We do not have evidence for a human influence.

(ii) I made the point that it was not appropriate to discount the radiocarbon dates on charcoal, which might be a few millennia older than the sediment, but not more. The authors responded that the charcoal dates were discounted because they have applied the quality rating scale of Rodrigues-Rey et al (2015), and the dates are 'b' quality and they only used a and a*. This is technically correct, but my point was how much older could the charcoal dates reasonably be than the age of the sediment (and the fossils), and the answer is a few millennia maximum and probably less. The authors then say the charcoal was 'badly preserved', and again, fine, but that would make the ages more likely minimum ages, and the dates do provide a constraint that suggest probably not younger than 40 and possibly quite a bit older. The 'reworked' argument I think is weak, though not entirely discountable I guess.

The exact magnitude of the age offset for charcoal entrained within a fluvial setting will be both site- and sample-specific, and there is no way of directly constraining this inaccuracy on a sample-by-sample basis *ex post facto*. Rather than speculate on the degree of bias affecting the reworked charcoal samples, we prefer a more objective approach – i.e., basing our final chronological interpretations on a formalised quality rating scheme, and using only those ages considered reliable on methodological and stratigraphic grounds, as set out by Rodriguez-Rey et al.. The potential ambiguities that could result from incorporating sub-standard charcoal data are reinforced by the reviewer comments. The charcoal ages could be interpreted as 'maximum' ages because the material has been reworked through the fluvial system, and/or they could be interpreted as 'minimum' ages because the charcoal was poorly preserved. The reviewer admits that both these opposing inaccuracies cannot be discounted, which essentially precludes reliable interpretation of the charcoal ages.

(iii) I also note, on reflection, that the Gledswood dates I referred to are unpublished I think, so no reason for the authors to have known about them. I do think its not credible to claim that people weren't around in the catchment – my view is that people were everywhere by then, but again that's only my opinion.

This opinion contrasts with Reviewer #4 who suggests that people were "pretty thin on the ground". We try to refrain from opinion regarding human presence. However, in our updated text we suggest that future concentrated field effort may establish an archaeological record older than 40 ka for the region. This is as far as we can go until direct evidence is found. There is no direct evidence of megafauna extirpation at South Walker Creek and no local or regional record confirming a human influence on megafauna.

Reviewer #3 (Remarks to the Author):

We thank Reviewer #3 for their comments.

I am happy with the authors' responses and changes to the MS. I think it's good to go.

Reviewer #4 (Remarks to the Author):

We thank Reviewer #4 for their additional comments and provide our comments below.

I have read both the revised manuscript and the SI. The increased clarification by changes in the text is very helpful. With the excuse that the authors could not find the Mead and Metzler reference I would advise they widen their search and perhaps refer to the other Mead and Meltzer reference from the same year. A quick google search will do it and then a visit to the library.

We have subsequently found a relevant reference and used this.

I am still critical of the treatment of the only two sites where megafauna and a record of humans co-occurring in Sahul have been identified: Nombe Rockshelter and Cuddie Springs. With Nombe, the recent dating project has confirmed the ages of the Unit where the megafauna were found.

It is interesting that the human record is found with fragmented faunal remains. Nonetheless when an interaction between humans and megafauna cannot be demonstrated, it does indicate that megafauna survived in the highlands till quite late. There is no excuse for ignoring these results. The evidence is clear.

This is incorrect. Although the recent dating project “indicate reasonable chronostratigraphic integrity for major strata” the authors were careful to stress this was “in one portion of the site” and that “the new dating program did not target the key contexts to address these issues”. Denham, T. and Mountain, M.J., 2016. Resolving some chronological problems at Nombe rock shelter in the highlands of Papua New Guinea. *Archaeology in Oceania*, 51(S1), pp.73-83.

If an updated chronostratigraphic assessment is published with direct relevance to the megafauna-bearing units, then this site would represent the northern-most reliably-dated megafauna site. We look forward to this occurring. If such results do convey reliably young ages, this would not undermine our interpretation for the extinction of megafauna at South Walker Creek. But until such data is confirmed and published we cannot use it.

Not everyone agrees that the Saltré et al dating confidence attributions are an accurate representation of the dating confidence. To dismiss the Cuddie Springs evidence as possibly disturbed is a clear indication that the authors have either not read the relevant papers or not understood them.

We have included Cuddie Springs within our regional assessment, including it in Figures and discussions; therefore, we have included it where possible. The literature remains contentious about Cuddie Springs and evidence for the site is not clear. Until resolved, we take a conservative approach to this site. We have presented our assessment of the published ages and presented our

interpretations based on the criteria used to rate particular dates and methods. We based this on the criteria established by Rodríguez-Rey et al. (2015). Our assessment was not based on Saltré et al. (2016). We differ with a more optimistic interpretation of the OSL data, rating it as a B-rating not a C, and the US-ESR an A-rating, not a B. Taking the FosSahul ratings for Cuddie Springs dates on face value we would have excluded the site entirely from our assessment. We see value in a criteria-based system such as this because it provides an avenue to undertake an independent assessment that can be corroborated or challenged equably.

All of this being said, our key interpretation that the extinction timing of megafauna within the MDB occurred at, or during, a major period of sustained hydroclimatic change would only be altered if the ages of all of the faunas within the MDB were found to be considerably older than the ranges presented here.

Furthermore, it is somewhat annoying to give ESR/U-series methods such an elevated position in the dating suite. How this has happened I am not sure, yet in other studies ESR-U series has been up to 10,000 years older than other dating (OSL/C14) methods. Are the authors now suggesting via Saltré et al that the Mungo deposits are 20,000 years older than the accepted OSL and C14, or that the Devils Lair sequence is also much older? Do they not see the consequences of promoting such interpretations because it fits their thesis? This is a rhetorical question of course.

Although references are not given by the reviewer, we are assuming that these particular examples from Mungo and Devil's Lair refer to Bowler et al. (2003) and Turney et al. (2001) respectively. These studies employed ESR dating on its own (i.e. not in combination with U-series dating), and therefore the resulting ages had been derived using assumed uranium uptake histories. This approach typically results in non-finite ages, and indeed these limitations are clearly stated in their associated publications, see Bowler et al. (2003, p840) "Reasons for the 20-kyr age discrepancy invite speculation. One factor may involve uncertainties in U-migration, which is enhanced in carbonate-rich sedimentary deposits such as those studied here. U-migration can considerably affect the accuracy of 'open system' ESR and U-series ages, such as those used to date the Mungo III skeletal remains"; and Turney et al (2001, p11): "Additional U-series analysis of the dental material would be required to refine further the ESR age estimates for this layer."

This is a completely different scenario to the ESR/U-series ages presented in the current study. By using ESR dating in conjunction with U-series mapping of the same specimens we are able to derive finite age estimates that take into consideration known, not assumed, U-series uptake history. In essence, the reviewer is comparing apples and oranges, and this concern does not take into account the physical principles of the dating techniques.

I would also suggest that the authors be more circumspect about the presence of humans. Until it is demonstrated (by empirical evidence) to be so then you cannot claim they were probably there. The work of Alan Williams has shown that the population levels across Sahul at this time were very low. Does that not indicate something? Furthermore, the fact there are only two megafauna sites in Sahul with an associated human record, which by the way these authors dismiss, does that also tell you that either megafauna or humans were pretty thin on the ground?

This comment is in complete opposition to the opinion of Reviewer #1 regarding the presence of humans at South Walker Creek. Regardless, neither our original or revised text has claimed a known presence of humans at South Walker Creek during the deposition of the fossil deposit. We originally stated that humans conceivably could have been present but in low numbers, using the work of Williams et al. (and others). During our first revision, to placate the opposing opinions by reviewers #1 and #4, we changed the text to read:

“With no evidence for a human presence within the FRB before ~19ka, or even more regionally before 40 ka, we cannot implicate people in the extinction of the SWC megafauna. Acknowledging the limited number of archaeological sites older than ~30 ka from eastern Sahul^{57,68} we cannot rule out that this absence is a result of significant under sampling, therefore, future concentrated field effort in eastern Sahul may establish an archaeological record older than 40 ka.”

We feel that this is a balanced response to differing opinions.

We assume the “only two megafauna sites in Sahul with an associated human record” referred to here are Cuddie Springs and Nombe Rock Shelter. We do not dismiss Cuddie Springs, as previously discussed above, and cannot include Nombe Rockshelter for reasons also discussed above.